# SynthBio: A Case Study in Human-AI Collaborative Curation of Text Datasets

**Ann Yuan**
Google Research
annyuan@google.com

**Daphne Ippolito**
Google Research
dei@google.com

**Vitaly Nikolaev**
Google Research
vitalyn@google.com

**Chris Callison-Burch**
University of Pennsylvania
ccb@cis.upenn.edu

**Andy Coenen**
Google Research
andycoenen@google.com

**Sebastian Gehrmann**
Google Research
gehrmann@google.com

## Abstract

NLP researchers need more, higher-quality text datasets. Human-labeled datasets are expensive to collect, while datasets collected via automatic retrieval from the web such as WikiBio [32] are noisy and can include undesired biases. Moreover, data sourced from the web is often included in datasets used to pretrain models, leading to inadvertent cross-contamination of training and test sets. In this work we introduce a novel method for efficient dataset curation: we use a large language model to provide seed generations to human raters, thereby changing dataset authoring from a writing task to an editing task. We use our method to curate SynthBio–a new evaluation set for WikiBio– composed of structured attribute lists describing fictional individuals, mapped to natural language biographies. We show that our dataset of fictional biographies is less noisy than WikiBio, and also more balanced with respect to gender and nationality.[1]

## 1 Introduction

Progress in machine learning depends on the availability of high-quality benchmark datasets, yet there is a shortage of such datasets for natural language generation. This is because collecting high-quality labeled text datasets is slow and expensive, requiring significant labor from skilled human annotators [35]. In addition, some tasks are sufficiently complex that the average crowd rater will be unable to perform them within a reasonable amount of time. Thus human-authored datasets typically consist of relatively short targets [52], and researchers often use automated retrieval-based methods, such as sourcing task examples from the internet, for dataset construction. However these methods can be problematic, as web text is noisy and biased. Moreover, with the growth in popularity of pre-trained language models, inadvertent cross-contamination of training and test sets is a real risk [15]. For these reasons, many in the machine learning community have begun to call for research into new dataset collection methods [9, 38].

In this work, we introduce a new method for efficient labeled text dataset curation that leverages the generative capabilities of the most recent large language models. We use a large language model similar in size to GPT-3 to generate a first draft of a dataset. Crowd raters modify these seed examples, which is a quicker task than authoring examples from scratch.

We test our method by curating SynthBio—a new benchmark dataset for WikiBio [32]. WikiBio is a structure-to-text task and dataset where the goal is to map from attribute lists about notable people (extracted from the structured data *infoboxes* present on their Wikipedia pages) to the first

---

[1] The dataset can be found at `https://storage.googleapis.com/gem-benchmark/SynthBio.json`.

35th Conference on Neural Information Processing Systems (NeurIPS 2021) Track on Datasets and Benchmarks.

Table 1: Properties of WikiBio and SynthBio. Columns # ref, # attrs, # words are the mean number of bios per infobox, attributes per infobox, words per bio, respectively. Div-2 is the number of unique 2-grams divided by the total number of 2-grams over all biographies (↑ is more lexically diverse). Ppl is perplexity of the bios according to GPT-2 (↓ is more fluent). The final column is the percentage of bios which use majority he/his (M), she/her (F), they/them (NB), or no majority (?) pronouns.

|                    | Inputs | Bios   | # Refs | # Attrs | # Words | Div-2 | Ppl  | % M/F/NB/?        |
|--------------------|--------|--------|--------|---------|---------|-------|------|-------------------|
| WikioBio Valid     | 72,831 | 72,831 | 1      | 12.47   | 97.7    | 0.21  | 19.5 | 60 / 12 / 2 / 27  |
| WikioBio Test      | 72,831 | 72,831 | 1      | 12.45   | 97.7    | 0.21  | 19.5 | 59 / 12 / 2 / 27  |
| SynthBio (unedited)| 2,793  | 8,304  | 3      | 21.30   | 234.0   | 0.12  | 4.78 | 45 / 40 / 9 / 6   |
| SynthBio (final)   | 2,249  | 4,692  | 2.1    | 19.57   | 112.0   | 0.22  | 16.2 | 38 / 37 / 23 / 2  |

paragraphs from those same pages (their *biographies*) [32]. Our synthesized benchmark consists of 2,249 infoboxes describing fictional persons, *each* mapped to an average of 2.1 biographies, for a total of 4,692 biographies (Table 1). We took advantage of the synthesis pipeline to showcase how datasets can be constructed with properties that deliberately differ from real world distributions. Notably, we include samples of individuals with common (e.g., scientist) as well as uncommon occupations (e.g., spy) (Table 3) and designed SynthBio to be more balanced with respect to gender and nationality compared to the original WikiBio dataset. Human evaluation shows that the biographies in SynthBio are also more faithful to their corresponding infoboxes, while just as fluent as those from the original dataset. Finally, we show that models trained on WikiBio do not perform as well on our dataset. This suggests SynthBio may be useful as a challenge set for evaluating models' ability to perform along the entire target distribution and to generate grounded text without benefit from real-world knowledge memorized during pre-training.

## 2   SynthBio: A Synthesized Benchmark for WikiBio

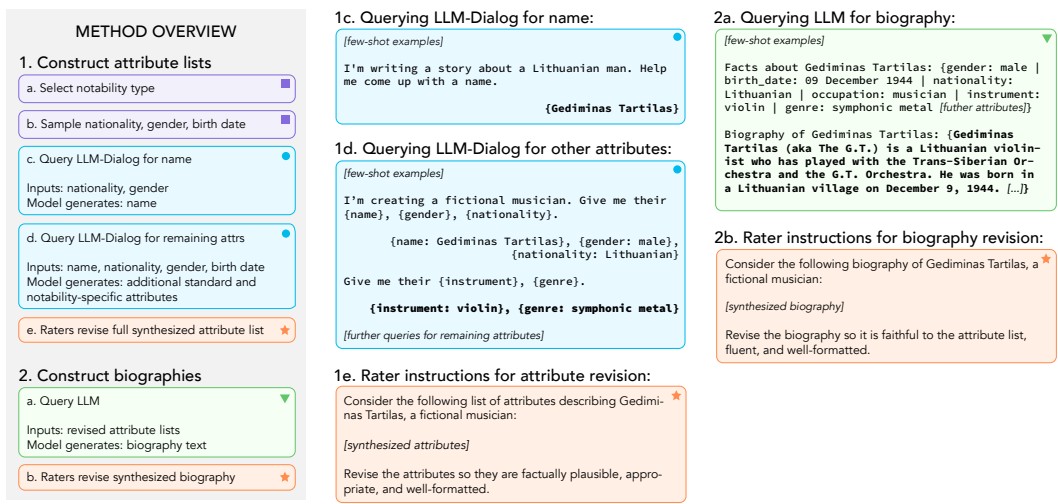

Figure 1: On the left, we show an overview of the steps involved in SynthBio's construction. On the right, we show examples of the prompts used for generation and the instructions given to raters. ■ are programmatic steps–selecting a notability type then randomly sampling basic attributes from a schema. ● are steps that query the LLM-DIALOG. ★ are steps that involve human raters. ▼ are steps that query the LLM. Text that is generated by a language model is **bolded**.

**WikiBio**   The WikiBio dataset consists of 728,321 infoboxes describing notable individuals, mapped to biographies written in natural English about those individuals. Annotated corpora for structure-to-text tasks are scarce, and WikiBio is a popular dataset to train models to translate structured data (infoboxes) into text (biographies). However, the dataset has several problems which make it difficult

Table 2: When primed with a few-shot prompt containing three biographies (but no attributes) and the name of a real individual in the WikiBio validation set, LLM is able to generate some accurate information. All text except the name in bold was generated. Green is info that can be found in the corresponding Wikipedia page's attribute box, while blue is info that is accurate but not present in the attribute box.

| Notability | Generated Bio |
|---|---|
| Author | **Ken MacLeod** is a Scottish writer, working in multiple genres. MacLeod began writing science fiction in 1997, with "The Stone Canal" – a nominee for a Nebula Award, Campbell Award, and BSFA Award. In 1999, MacLeod was awarded the Campbell Award for his novel "Newton's Wake" as well as the 2001 BSFA Award for 'The Cassini Division'. |
| Sportsperson | **Bill Kazmaier** is an American professional Powerlifter, strongman, and Weightlifter. During Kazmaiers career they held multiple world records across various strongman and powerlifting federations. Kazmaier began training in 1976, and competed until 1990 with multiple injuries. |
| Musician | **Al Alberts** was born on July 22, 1922, in Philadelphia, Pennsylvania. He learned piano as a child and became a vocalist in the late 1940s. He became most famous as the founder of The Four Aces. |

to tell whether a high-performing model on WikiBio has truly learned the structure-to-text task. First, it is noisy—biographies often contain information that cannot be found in the corresponding infobox, and vice versa. In addition, many infoboxes are either very short or indecipherable due to artifacts from improper formatting. As argued by Søgaard et al. [48], using an i.i.d. test split will include this noise in any evaluation results and thus distract from the signal.

Second, the validation and test sets are formed from real people. For well-known figures, there may be many documents on the internet which describe this person. It is known that pre-trained models are able to memorize facts from their training data [44], and we observed qualitatively that large language models such as GPT-3 are capable of generating factually correct summaries for famous figures without being given attributes describing those figures in the input. A few examples are shown in Table 2. Moreover, real data contains biases such as the fact that mathematicians are more often men or that most biographies on English Wikipedia are of Americans and Europeans [54, 25, 18]. Machine learning methods trained on biased data may perpetuate these biases in undesirable ways [31, 60].

| Type | WikiBio % | SynthBio % |
|---|---|---|
| musical artist | 11.7% | 12.5% |
| sportsperson | 9% | 12.5% |
| scientist | 4.4% | 12.5% |
| writer | 3.6% | 12.5% |
| artist | 2.5% | 12.5% |
| spy | 0.03% | 12.5% |
| theologian | 0.03% | 12.5% |
| mountaineer | 0.009% | 12.5% |

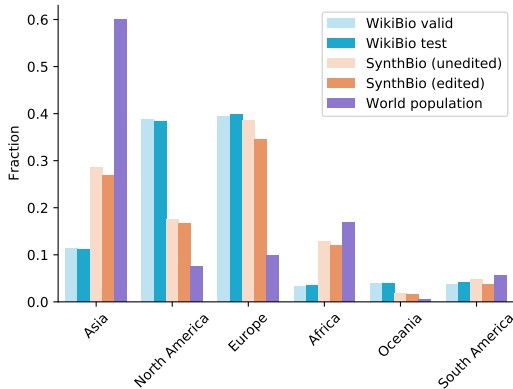

Table 3: Types of individuals included in Synth-Bio, along with their respective representations in WikiBio and SynthBio.

Figure 2: The distribution of continents for the "birth place" attribute compared to the distribution of world population [2]. For 23% of WikiBio and 8% of SynthBio infoboxes, continent information was unavailable either because the attribute was missing or the Geocoding API failed to parse it.

Table 4: (top) Human evaluation results on the original WikiBio dataset versus SynthBio (see Appendix 8.4 for inter-annotator agreement scores). (middle / bottom) Human evaluation results on the outputs of model trained on WikiBio. Coverage is computed as (number of recalled attributes) / (number of attributes). Faithfulness scores are reported on a scale of *Faithful* - 4 (■), *Mostly faithful* - 3 (■), *Somewhat faithful* - 2 (■), and *Not faithful* - 1 (■). Fluency is computed as (number of biographies deemed fluent) / (number of biographies).

| Dataset | Coverage | Faithfulness | | Fluency |
|---|---|---|---|---|
| WikiBio | 0.44±0.007 | μ = 2.5 | | 0.97 |
| SynthBio | 0.86±0.006 | μ = 3.75 | | 0.97 |
| T5 (783M) beam - WikiBio | 0.35±0.006 | μ = 3.16 | | 0.99 |
| T5 (783M) beam - SynthBio | 0.33±0.003 | μ = 3.7 | | 0.99 |
| T5 (783M) sampled - WikiBio | 0.38±0.007 | μ = 2.3 | | 0.99 |
| T5 (783M) sampled - SynthBio | 0.37±0.004 | μ = 2.5 | | 0.98 |

**SynthBio** By constructing a synthetic dataset, it is possible to control for such biases and reduce noise. Given the challenges surrounding the WikiBio dataset, we set out to curate an evaluation dataset that would address some of the issues present in the original validation and test sets. SynthBio, our WikiBio benchmark dataset, consists of 2,249 infoboxes describing fictional individuals and 4,692 biographies (see Appendix Table 8 for samples from SynthBio). Models cannot achieve high-performance on our benchmark by simply memorizing facts about real-world individuals. SynthBio includes both common (e.g., musician) and uncommon (e.g., mountaineer) notability types, thus enabling evaluation on language phenomena that may be atypical in the WikiBio dataset (Table 3). We attempted to include equal representation among male, female, and non-binary individuals - Table 1 shows that SynthBio has a much more evenly distributed use of pronouns than WikiBio. The fictional individuals also hail from a wider variety of countries than individuals in the WikiBio dataset (Figure 2) (however the distributions are not perfectly uniform - due to certain synthesized examples being deemed unsalvageable by annotators and thus not included in the final dataset).

Unlike in WikiBio, each infobox in our dataset maps to multiple biographies (Table 1). There are many ways to express a structured attribute list in natural language, thus this feature of our dataset also helps to provide a more accurate evaluation of structure-to-text task performance. Alva-Manchego et al. [4] demonstrate the importance of curating NLG datasets with multiple references when trying to train models to perform generation tasks with many acceptable outputs.

Lastly, since human annotators revise each synthesized example, SynthBio demonstrates higher faithfulness between the input infoboxes and the target biographies, enabling more accurate evaluation of structure-to-text task performance (Table 4).

## 3 Dataset curation methodology

In this work we propose a hybrid workflow in which a language model generates a first draft of a dataset example, and a human edits that draft. Based on evidence that editing is faster than writing [57], this workflow allows us to collect examples more efficiently than through a purely human workflow, while ensuring that the examples are of a higher quality than those produced by a fully automated workflow.

We use two language models for different parts of the pipeline. They are both 137B parameter, dense left-to-right decoder-only Transformer models, similar to GPT-3. The model we refer to as LLM was trained on 1.95B documents extracted from web sites, which were tokenized into 2.49T BPE tokens with a SentencePiece [29] vocabulary of 32K tokens. This model was then finetuned on a curated, high-quality subset of the data that was identified to be in a conversational format. We refer to this second fine-tuned model as LLM-DIALOG.

Our hybrid curation pipeline works as follows (see Figure 1): (1) a language model synthesizes infoboxes, (2) humans revise and perform quality control over synthetic infoboxes, (3) a language model synthesizes biographies based on the revised infoboxes, (4) humans revise and perform

Table 5: Number of biographies and attributes lists synthesized, discarded, and edited (top), similarity between the original synthesized and final edited versions (middle), and human cost of editing (bottom). Edit distance is character-level and for attributes is computed on the serialized attribute string.

| Property | Attributes | Biographies |
|---|---|---|
| Num synthesized | 2,793 | 8,304 |
| Num unsalvageable | 64 (2.3%) | 3,612 (43%) |
| Num edited | 1,216 (44%) | 4,555 (55%) |
| Avg edit dist | 34 | 560 |
| Self-ROUGE-l | N/A | 0.60 |
| Self-ROUGE-2 | N/A | 0.45 |
| Num raters | 6 | 11 |
| Avg time spent | 154s | 239s |

quality control over synthetic biographies. Such a back-and-forth workflow has previously been characterized as an effective way to take advantage of the unique qualities of the model and the human in a collaborative setting [21]. Indeed we observed that human raters performed many small (in terms of edit distance) but high-impact edits on the generated dataset, for example changing pronouns to match the gender indicated by the infobox (see Appendix 8.3 for additional examples of specific edits made). Such an edit can be made quickly (the error is easily identified, and only a few characters need to be changed), while it significantly improves the faithfulness of the biography.

Now we describe each step in detail. For steps 2 and 4, we include the exact instructions provided to raters in the supplementary materials, in which we describe how to adjudicate edge cases.

## 3.1 Step 1: Synthesizing Attribute Lists

To synthesize attribute lists, we use a combination of templated attributes, language model-generated attributes, and human revision. We started by generating 2,800 attribute lists. Annotators then revised and filtered these attribute lists, resulting in a total of 2,793 attribute lists.

**Programmatically selected attributes** First, a notability type was selected for each individual. Table 3 lists the eight notability types included in our dataset. We chose a mixture of common and rare notabilities. *Musical artist, sportsperson, scientist, writer, artist* are all among the 10 most common notabilities on Wikipedia, while the types *mountaineer, spy, theologian* are among the 20 least common (see the Appendix for a more detailed discussion of our selection process). In total, we generated 350 attribute lists for each of the eight notability types.

Next, for each individual, attributes whose distribution we wanted to precisely control were selected from a template. These included gender, nationality, and birth date. *Nationality* was sampled between 5 randomly selected countries for each notability type. *Birth date* was uniformly sampled between 1850 and 2000, a date range that was chosen to make factual plausibility checking easier for human annotators (see Section 3.1). *Gender* was uniformly sampled between *male*, *female*, and *non-binary*. While *birth date* and *nationality* are part of the standard infobox scheme, *gender* is not, but was included as a proxy for a person's pronouns to meet the goal of reducing gender bias in SynthBio.

**LM Generation of additional attributes** Additional attributes were generated with LLM-DIALOG. This included a mixture of common attributes (*birth place, death date, death place, death cause, resting place, partner, children, mother, father*) and notability-specific attributes defined by the notability's schema (Appendix - Table 9). The first attribute we generate is a name. This is done by prompting the model with a request for name suggestions for a fictional person (Figure 1 - 1b). To generate the additional attributes, we then prompt the model with a staged conversation in which we ask for an individual's birth place, death date, and death place, and the model responds with those details (Figure 1 - 1c). Subsequently we ask the model for the remaining attributes in groups of two or three, appending its previous responses to the prompt at every turn. We chose to use LLM-DIALOG for attribute generation rather than LLM because an exchange consisting of requests for information and responses containing that information readily fits the conversational format. For all steps of attribute generation, the conversational context includes few-shot examples manifested as previous conversational turns in the same format as we would like the model to generate.

**Human revision of infoboxes** We then asked human raters to evaluate and revise the synthetic infoboxes according to the following criteria:

*Factual plausibility:* Though the people in SynthBio do not exist, we ask raters to ensure that they are factually plausible according to two constraints. First, the person ought to be able to exist without changes to physical laws (e.g., a person's death date may not precede their birth date) and major historical events (e.g., no saxophonists before the invention of the saxophone in the 1840s). Second, though infoboxes can include fictional proper names (e.g., names of specific persons, places, etc), it should not include fictional common nouns (e.g., an invented name for a nonexistent instrument).

*Appropriateness:* We asked raters to ensure infoboxes are written succinctly and follow conventions for the various fields as described in the official Wikipedia infobox schemas for each notability type.[2]

*Formatting:* We asked raters to ensure infoboxes are properly formatted and do not contain artifacts from generation.

Raters spent an average of 154 seconds reviewing each infobox, resulting in 2,736 infoboxes (2% were deemed unsalvageable) (Table 5). 44% of infoboxes were edited to some degree, confirming the importance of including humans in the curation workflow. A single annotator was assigned to each item, thus we did not consider inter-annotator agreement. However, to ensure adherence to our annotation guidelines, each annotator went through a training phase during which the authors provided feedback on their work.

### 3.2   Step 2: Synthesizing Biographies

**LM generation of biographies**   For each of the 2,736 revised infoboxes we used LLM to generate three target biographies (8,208 biographies total). We prompted LLM with the infobox followed by the the text `Biography of [name]:  {"` (Figure 1 - 2a), and the model was made to continue generating tokens until it produced a `"}`, indicating the end of a biography. To improve output quality, we prepended three few-shot examples in the same format as the goal example to the prompt. All few-shot examples (see Appendix 8.2) were written by the authors and described fictional individuals with the same notability type as the individual whose biography LLM was being asked to generate. These few-shot examples are included in the final dataset.

**Human revision of biographies**   We then asked human raters to evaluate and revise the synthesized biographies according to the following criteria:

*Faithfulness:* A biography is faithful to its corresponding infobox if it only contains information that can be inferred from the infobox. We asked raters to delete text from the biographies that was not attributable to the infobox, or revise text that reflected the infobox with minor inaccuracies. We also asked raters to edit the biographies in order to fill informational gaps, for example in case an infobox lists a person's death date but the date is not mentioned in the biography.

*Fluency:* We asked raters to edit the biographies for disfluencies and repetitive language.

*Formatting:* We asked raters to remove generation artifacts from the biographies.

Raters spent an average of 239 seconds reviewing each biography, resulting in 4,692 biographies (43% were deemed unsalvageable), of which 4,555 were edited (Table 5). The paper authors performed a final proofreading of the dataset.

## 4   Dataset evaluation

Our final dataset consists of 2,249 infoboxes and 4,692 biographies. In this section we discuss results from evaluating model performance on the final dataset.

**Automated evaluation**   As baseline models, we investigate T5 [41] across three sizes ranging from 77M to 783M parameters. Each model was finetuned for 10,000 steps on the WikiBio training data on an internal cluster using TPUv3 chips using the process described by Kale and Rastogi [27]. We report results with beam search inference with a beam size of 4 and with top-k sampling with k=40 and a temperature of 1.0 [16]. For the sampling-based approach, we run inference five times and report average and standard deviation of the results.

---

[2]https://en.wikipedia.org/wiki/Category:Biographical_templates_usable_as_a_module - last accessed August 27, 2021

Table 6: Evaluation results of model outputs on the original WikiBio test set, the unedited, and the final version of SynthBio. For sampled outputs, we report the average of scores across five generations for each system and data point. The standard deviation across sampled outputs for all metrics is <0.01 (<0.1 for ROUGE).

| Model | PARENT-P | PARENT-R | PARENT-F | BLEURT | BLEURT-20 | ROUGE |
|---|---|---|---|---|---|---|
| **WikiBio** | | | | | | |
| T5 (77M) beam | 0.845 | 0.104 | 0.104 | -0.225 | 0.468 | 40.0 |
| T5 (248M) beam | 0.841 | 0.106 | 0.106 | -0.204 | 0.476 | 41.2 |
| T5 (783M) beam | **0.850** | 0.107 | 0.107 | **-0.190** | **0.480** | **41.9** |
| T5 (77M) sampled | 0.604 | 0.114 | 0.163 | -1.035 | 0.191 | 33.2 |
| T5 (248M) sampled | 0.604 | **0.115** | 0.164 | -1.018 | 0.181 | 33.8 |
| T5 (783M) sampled | 0.622 | 0.114 | **0.166** | -1.000 | 0.177 | 35.0 |
| **SynthBio (unedited)** | | | | | | |
| T5 (77M) beam | 0.711 | 0.029 | 0.029 | -0.596 | 0.372 | 16.7 |
| T5 (248M) beam | 0.718 | 0.029 | 0.029 | -0.583 | 0.376 | 17.3 |
| T5 (783M) beam | **0.721** | 0.029 | 0.029 | **-0.563** | **0.378** | 17.4 |
| T5 (77M) sampled | 0.560 | **0.031** | **0.052** | -0.715 | 0.315 | 19.8 |
| T5 (248M) sampled | 0.559 | **0.031** | **0.052** | -0.655 | 0.313 | 20.0 |
| T5 (783M) sampled | 0.571 | 0.030 | **0.052** | -0.575 | 0.326 | **20.2** |
| **SynthBio (final)** | | | | | | |
| T5 (77M) beam | 0.729 | 0.027 | 0.027 | -0.603 | 0.358 | 19.7 |
| T5 (248M) beam | 0.735 | 0.027 | 0.027 | -0.584 | 0.362 | 20.2 |
| T5 (783M) beam | **0.736** | 0.027 | 0.027 | -0.567 | **0.364** | 20.4 |
| T5 (77M) sampled | 0.553 | 0.028 | **0.049** | -0.676 | 0.326 | 22.3 |
| T5 (248M) sampled | 0.551 | **0.029** | **0.049** | -0.614 | 0.316 | 22.4 |
| T5 (783M) sampled | 0.562 | 0.028 | **0.049** | **-0.538** | 0.326 | **22.6** |

We report three metrics typically used to estimate output quality in data-to-text tasks. (1) PARENT [14] is a metric that measures the overlap between attribute values in the input and the generated output with respect to a reference. It provides a precision, a recall, and an F-score. (2) BLEURT [46] is a learned metric trained to measure semantic similarity between a reference and a generated output. In addition to the original metric we also report the newer version BLEURT-20 [47, 39] (3) ROUGE [33] measures the lexical overlap between reference and generation. We report ROUGE-L which measures the longest common subsequence between reference and generation. Since our dataset has multiple references, we average the scores on all references for each example for PARENT and BLEURT and use a multi-reference implementation of ROUGE.

**Human evaluation** We asked crowd raters to evaluate 400 samples from SynthBio and WikiBio, as well as 400 beam search and sampled outputs from the best performing model T5 (783M), according to the following criteria:

*Coverage:* This refers to how many of the attributes listed in the infobox are mentioned in the biography. To evaluate, crowd raters were presented with a biography and corresponding infobox, and were asked to check off the attributes mentioned in the biography.

*Faithfulness:* A biography is faithful to an infobox if it includes *only* information that can be found in the infobox. Crowd raters evaluate faithfulness on a scale of *Faithful* - nearly every piece of information in the biography is in the infobox, *Mostly faithful* - more than half of the information is in the infobox, *Somewhat faithful* - less than half of the information is in the infobox, and *Not faithful* - almost none of the information is found in the infobox.

*Fluency:* Crowd raters were asked to indicate their agreement with the statement "The biography is clear, natural, and syntactically well-formed."

**Results** The human evaluation results presented in Table 4 indicate that SynthBio biographies demonstrate much greater coverage and faithfulness than WikiBio biographies, while being just as fluent. Thus models that successfully learned the biography-generation task should perform better on

SynthBio than WikiBio, due to the relative absence of unpredictable noise in SynthBio references. Contrary to this expectation, models obtain lower precision scores on SynthBio than WikiBio as measured by PARENT-P and ROUGE, shown in Table 6. This can have multiple reasons: (1) the focus of SynthBio on tail-examples with less representative and more highly specified attributes, (2) the fact that the model cannot rely on memorization, and (3) the effect of model-produced references combined with the rewriting by speakers of a dialect different from the original data.

Focusing on point (1), we further investigated performance on subpopulations. We observe no significant difference in performance as a function of number of attributes in the input except for outlier cases where inputs are extremely long (>250 words) or short (<15 words). Similarly, performance is equal across genders. There is a higher variance among the results across nationalities (e.g., ranging from 0.028 to 0.084 PARENT-F for T5 (783M)), but there is no discernible pattern concerning which nationalities do better or worse. Finally, we investigated performance by notability type. Since we selected both common and exotic types, we hypothesized that models would perform better on common types. This is also not the case – "Sportsperson" is the only type for which the model performs significantly better than the others. Points (2) and (3) are assessed through our human evaluation which is reference-less and has no rater overlap with the annotators. It thus evaluates whether model outputs by themselves are better for one of the datasets. Surprisingly, we observe the opposite result as in the automatic evaluation, shown in Table 4. All results show a high degree of fluency and low coverage while faithfulness differs significantly. Faithfulness on SynthBio with beam search is significantly higher than that for WikiBio. The results further disagree with the PARENT-F results which showed a significant increase in the sampled results, possibly due to spurious overlaps between sampled words and the infobox.

However, these results rule out explanation (2) and (3) since we would have seen equal or better results for WikiBio. This leaves us with a final possible explanation – one that was shown by Freitag et al. [20] for machine translation. Since low-quality references lead to inflated automatic evaluation scores, the WikiBio results are artificially higher than they should be. Consequently, researchers may use SynthBio to more accurately characterize models' performance on the biography generation task along the entire target distribution.

When comparing the final and the unedited version of SynthBio, the differences between the two are less substantial than those compared to WikiBio. Since most of the edits were minor fixes, the overall ranking of models is not changed drastically. However, we can observe a drop in PARENT scores in the final version, indicating that the addition of originally dropped attributes into the reference is helping make those scores more precise. We also observe a corresponding decrease in BLEURT-20. The increase in ROUGE can be explained by the increased length of references, which is more likely to match unrelated phrases in the produced text.

# 5 Related work

**Curating structure-to-text datasets**    Many existing structure-to-text datasets are curated via automatic retrieval from the web, such as WikiBio and RotoWire [56]. Others such E2E [36] are created by crowdworkers asked to compose natural language references corresponding to structured inputs. A hybrid approach was taken in the case of ToTTo [37], a table-to-text dataset, for which researchers first retrieved samples from Wikipedia, then asked crowdworkers to edit those references. The Schema Guided Dialogue Dataset [42] is another recently introduced dataset curated via a hybrid approach in which crowd raters were assisted by a templating system in composing their references. We were inspired by this approach in creating SynthBio, although we use a generative language model in place of a templating system to improve lexical diversity.

**Synthetic dataset creation and augmentation**    With recent advances in generative language modeling there has been growing interest in leveraging their capabilities for dataset synthesis and augmentation [6]. [40] achieve state-of-the-art results on a question answering model trained only on synthetic data generated by GPT-2. [12] use GPT-3 to augment a small dataset of human-labeled medical dialogues and train improved summarization models on the resulting dataset. [11] achieve new state-of-the-art results on E2E and WebNLG benchmarks by using GPT-2 to augment training data. There has also been work in using synthesized datasets for counterfactual generation [57], causal inference evaluation [58], clinical entity recognition [26], and more [8, 3, 5].

**Human-model collaborative dataset creation**   However there has been less work exploring how this latest generation of language models might help human raters in a collaborative dataset curation workflow, in which models assist with curation but humans have final say over the output. Such hybrid workflows have typically been studied with automatic systems whose role is not to generate text, but rather to label or to retrieve text. Snorkel [43], for example, is a platform enabling experts to write labeling functions for generating datasets. [52] explores the possibility of collecting natural language inference data by automatically retrieving premise-hypothesis sentence pairs from web documents, which annotators then simply label. However the approach was deemed inferior to a fully manual approach due to data quality issues with the synthetic dataset. Finally, Fanton et al. [17] propose a data collection method where a language model is iteratively finetuned on human-edited versions of its own generations.

GitHub's recent Copilot project[3] showed the possibility of using large language models like GPT-3 to scale and enhance human capabilities, rather than to replace them. This follows a rich line of research in human-model collaborative interfaces, in which models generate draft outputs that humans edit [22, 10, 28, 13, 24, 23].

# 6   Discussion and broader impact

**The argument for curating datasets**   Whether or not datasets should be curated to alter underlying distributions is a foundational issue. Rogers [45] summarize the arguments for and against curation that followed after the publication of work by Bender et al. [7] which argued strongly for curation. One of the core questions is whether we should study the world as it is or the world as we want it to be, where "world" refers to extant sources of data, such as Wikipedia. Wikipedia's editors are over 90% male[4], and as discussed earlier, the people represented in WikiBio heavily skew toward male North-Americans with a small set of notability types. Our paper takes the stance that in addition to evaluating on the world as it is, researchers benefit from having the option to evaluate their models on a more uniform distribution of the population. Synthesizing novel datasets is one technique that serves this goal.

**Advantages of synthetic data**   The synthesis of novel data which still retains the fluency of real-world data but has more controllable bias and reduced hallucination may be a powerful tool for evaluating the capacity of language models, especially as they approach real-world deployment. For example, there are increasingly urgent calls to evaluate modes' capacity for grounded, faithful generation, yet there are insufficient tools for this evaluation. Structure-to-text tasks can serve this purpose, but since most existing evaluation datasets are created by automated retrieval-based methods, they themselves contain hallucinations. Thus, researchers cannot easily use datasets like WikiBio to evaluate their models' tendency to hallucinate, as their models may have simply learned noise present in the training data. In addition, undesirable bias in real-world data, especially with respect to underrepresented groups, can be controlled in synthetic data, enabling evaluation of model performance on comparatively rare language phenomena. We show that using our approach it is possible to create a dataset that is grounded and faithful, and more inclusive of underrepresented groups. We encourage more researchers to create synthetic datasets which measure complementary properties to what can be evaluated with existing datasets.

**Validity of using language models to generate benchmark data**   Underlying the presented work is the question of whether we should use generative language models as part of a data synthesis pipeline. This question is especially important in light of the many flaws and potentially harmful biases exhibited by large language models [30, 55, 50, 59, 53, among others]. We strongly argue against fully autonomous corpus creation which would suffer from the aforementioned biases. The fact that our dataset is human-revised is crucial as we do not want the model to generate toxic, discriminating, or otherwise undesirable text. As shown in Figure 1, we constrained the generation to heavily supervised prompts, and had human evaluators quality control every sample in our dataset while performing quick and easy but high-impact edits to produce the final output. As a result, the differences between the models' initial outputs and our final released dataset are substantial (Table 5).

---

[3]Copilot: `https://copilot.github.com/` - last accessed August 24, 2021
[4]`https://en.wikipedia.org/wiki/Wikipedia:Gender_bias_and_editing_on_Wikipedia`

However, revision by human annotators is unlikely to resolve more subtle biases and correlations that are present in text generated by large language models. For instance, our dataset more commonly shows birthplaces that are not related to a person's nationality than the original WikiBio. There is also an overrepresentation of Western institutions, awards, and attributes throughout the biographies. As a specific example, the synthesized individual Seung Ki Ahn's nationality is Korean, yet he is described as serving in the US military. Additionally, 5.1% of SynthBio individuals were listed as having a PhD in their education attribute, compared to 2% of US population [1], and even fewer for the global population. In practice, it is difficult to intentionally introduce some discrepancies from real-world distributions (e.g., balancing genders for each notability type) without unintentionally introducing others, especially subtle ones. For example, biographies often showed inconsistencies between a subject's gender identity and their pronouns, swapping between they/them pronouns and she/her or he/him pronouns. This was especially prevalent for non-binary subjects.

We note that all these inconsistencies were observed in the dataset even after it was audited by human raters specifically instructed to correct pronoun inconsistencies, suggesting stricter oversight may be required to address this issue in future data collection work. Thus, there is still much future work to do on enabling more fine-grained control over the data seeding and generation process. We believe human-AI collaboration to be a promising direction for the controlled curation of datasets which should not replace, but can be used in addition to, other curation methods.

For further discussion of these issues, please see the data card for SynthBio in the Appendix.

**Use of human raters**    Lastly, we note that for any dataset created fully or in part by human workers, the final dataset is only as good as the skills of the humans involved. Our method does not eliminate the need for finding or training skilled annotators. However it may reduce it, as revising examples is an easier task than constructing examples from scratch, especially for unintuitive, highly structured tasks like WikiBio.

Development of dataset construction methods that lower the workload for human annotators and play to their strengths is a promising direction for more investigation. Open research questions abound: How much faster is revision than generation for different tasks? For structure-to-text tasks in particular, is it easier to pare down text containing many hallucinations, or to add missing details to simple text?

# 7   Conclusion

In this work we introduced a workflow in which human raters collaborate with a large language model to efficiently curate a high-quality labeled text dataset. We presented a case study of this workflow in curating a new structure-to-text dataset, SynthBio, consisting of infoboxes and biographies describing fictional individuals that can be used as an evaluation set for models trained on the WikiBio dataset. We presented both human and automatic evaluations of our dataset to show that it is less noisy than WikiBio, and also more balanced with respect to gender and nationality.

The results from our case study suggest it is worth investigating the potential of collaborative human-AI data curation workflows for other types of NLP tasks, for example annotation. However, further work is still needed to rigorously evaluate the cost and quality tradeoffs for hybrid, purely synthetic, and purely manual dataset construction workflows. We look forward to seeing more work that uses a combination of automatic language generation and human revision to construct novel datasets.

Another potential use case of the type of dataset constructed in this work is as a final finetuning set. While we only evaluated SynthBio as an evaluation set, prior work by Solaiman and Dennison [49] demonstrated that small, highly curated datasets can be used during a finetuning stage to address issues with model bias.

## Contributions

- Ann Yuan designed the dataset synthesis pipeline, ran all inference with LLM and LLM-DIALOG, developed the interfaces used for human revision, shepherded the human revision process, and contributed to paper writing.

- Daphne Ippolito measured the properties of WikiBio and SynthBio, analyzed the tendencies of LLM to memorize information on real people, gave feedback on the synthesis pipeline and the human revision interfaces, and contributed to paper writing.

- Vitaly Nikolaev gave feedback on the synthesis pipeline and the human revision interfaces.

- Chris Callison-Burch offered mentorship and contributed to paper writing.

- Andy Coenen gave feedback on the human revision interfaces and the paper draft.

- Sebastian Gehrmann trained models on the WikiBio task, ran automatic evaluation on the SynthBio and WikiBio evaluation sets, gave feedback on the synthesis pipeline and the human revision interfaces, and contributed to paper writing.

## Acknowledgments and Disclosure of Funding

We thank Lucas Dixon, Meredith Morris, James Wexler, and Martin Wattenberg for providing feedback on our manuscript. We thank Kendra Schultz, Mike Green, and Qazi Rashid for their work conducting a thorough fairness audit of SynthBio.

Chris Callison-Burch and Daphne Ippolito's research is supported in part by the DARPA KAIROS Program (contract FA8750-19-2-1004), the DARPA LwLL Program (contract FA8750-19-2-0201), and the IARPA BETTER Program (contract 2019-19051600004). The views and conclusions contained herein are those of the authors and should not be interpreted as necessarily representing the official policies, either expressed or implied, of DARPA, IARPA, or the U.S. Government.

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

# 8 Appendix

## 8.1 Choosing Notability Types

Table 7 contains a list of all of the notability types that have attribute templates, sorted by the number of pages. These were extracted from `https://en.wikipedia.org/wiki/Category: People_and_person_infobox_templates`. As can be seen, there is high variability on the level of specificity in these notability types.

For SynthBio, we aimed to select notability types that were at roughly similar levels of specificity, and to have samples from common as well as exotic notability types. The types *musical artist, sportsperson, scientist, writer, artist* are all among the 10 most common types. We excluded officeholder (the most common type) because it is a complex infobox type with different attributes depending on the type (president, prime minister, member of Indian legislative body, etc.). Also officeholders would be particularly susceptible to factual plausibility issues. We also excluded military person (the fifth most common type) because many attributes only make sense given certain values of other attributes. We also excluded cricketer, baseball biography, NFL biography (7th, 8th, 9th most common) because they are all species of sportsperson. The types *mountaineer, spy, theologian* are among the 20 least common types in Wikipedia (excluding those with <20 pages). Many of the obscure types tend to have very few attributes, or to be a species of sportsperson, or to be simply problematically obscure (e.g., FBI Ten Most Wanted). After process of elimination we were left with mountaineer, spy, and theologian.

Table 7: All notability types in Wikipedia that have attribute templates, sorted by the number of pages that use them.

| |
|---|
| officeholder (180342) • musical artist (115630) • sportsperson (98534) • scientist (41728) • military person (39523) • writer (33973) • cricketer (30606) • baseball biography (26106) • NFL biography (25215) • artist (25063) • basketball biography (18863) • royalty (18820) • ice hockey player (18409) • Christian leader (15644) • cyclist (15329) • rugby biography (13710) • college coach (10597) • rugby league biography (9170) • swimmer (8950) • academic (8948) • tennis biography (8323) • noble (6512) • CFL biography (6077) • martial artist (5207) • volleyball biography (4966) • criminal (4630) • handball biography (4530) • saint (4446) • religious biography (4245) • figure skater (3973) • golfer (3957) • professional wrestler (3844) • comics creator (3393) • architect (3132) • philosopher (2974) • gymnast (2917) • badminton player (2789) • pageant titleholder (2627) • speed skater (2514) • skier (2446) • curler (2305) • model (2287) • comedian (1945) • YouTube personality (1934) • economist (1815) • medical person (1558) • college football player (1475) • sailor (1438) • field hockey player (1347) • horseracing personality (1152) • clergy (1097) • darts player (956) • engineer (934) • chef (889) • football official (881) • squash player (809) • Jewish leader (795) • table tennis player (772) • alpine ski racer (770) • astronaut (757) • poker player (756) • snooker player (725) • presenter (709) • dancer (634) • Latter Day Saint biography (623) • fencer (550) • sumo wrestler (545) • netball biography (499) • police officer (498) • lacrosse player (463) • amateur wrestler (442) • video game player (420) • War on terror detainee (385) • biathlete (358) • bodybuilder (335) • water polo biography (322) • Playboy Playmate (298) • classical composer (295) • aviator (290) • Native American leader (264) • go player (258) • spy (247) • pirate (245) • theologian (245) • climber (220) • surfer (197) • sport wrestler (191) • Twitch streamer (166) • pool player (162) • sports announcer details (154) • NCAA athlete (147) • GAA manager (108) • National Hockey League coach (106) • FBI Ten Most Wanted (103) • Magic: The Gathering player (103) • mountaineer (82) • professional bowler (57) • pelotari (18) • Instagram personality (6) • checkers biography (3) |

Table 8: Several final SynthBio dataset examples, after all human edits.

| Infobox | Biography |
|---|---|
| name: Tacettin Güntekin \| gender: male \| nationality: Turkish \| birthdate: 11 October 1930 \| birthplace: Istanbul \| deathdate: October 10, 1972 \| deathplace: Istanbul, Turkey \| deathcause: heart attack \| restingplace: Ankara \| almamater: Yale University, Oxford University \| education: PhD in Turkish literature \| occupation: professor, novelist \| notableworks: Saatleri Ayarlama Enstitüsü, Çalıkuşu \| language: Turkish \| genre: fiction, realism \| awards: Geschwister-Scholl-Preis (1983) \| mother: Zafer Güntekin \| father: Hüseyin Güntekin \| partner: Pınar Güntekin \| children: 2 boys, 1 girl | Tacettin Güntekin (11 October 1930 - 10 October 1972) was a Turkish professor and novelist, best known for his books "Saatleri Ayarlama Enstitüsü" and "Çalıkuşu". Born in Istanbul to Zafer and Hüseyin Güntekin, Tacettin Güntekin eventually attended Yale University and then Oxford University, where he obtained a PhD in Turkish literature. He died of a heart attack in 1972 in Istanbul and was laid to rest in Ankara. Güntekin was the 1963 recipient of the Geschwister-Scholl-Preis. Güntekin and his partner Pinar had three children. |
| name: Mari Rami \| gender: female \| nationality: Andorran \| birthdate: 28 January 1901 \| birthplace: Andorra, United States \| deathplace: U.K. \| deathcause: heart attack \| restingplace: private \| almamater: University of Andorra \| education: PhD in philosophy \| occupation: philosopher, novelist, poet, playwright \| notableworks: A Journey to Paradise \| language: Catalan and Spanish, French \| genre: historical fiction, philosophical fiction \| awards: Lletra d'Or \| mother: Marta Rami \| father: Enric \| partner: Emilia \| children: Tome and Maria | Mari Rami was an Andorran philosopher, novelist, poet, playwright best known for her novel A Journey to Paradise. Born on January 28, 1931 in Andorra to parents Marta and Enric, Rami attended the University of Andorra where she obtained a PhD in philosophy. Rami wrote historical and philosophical fiction in three languages: Catalan, Spanish, and French. Rami was a recipient of the Lletra d'Or. Rami and her partner Emilia had two children: Tome and Maria. She died in the U.K of a heart attack. Maria was laid to rest in a private place. |
| name: Alvaro Ochoa \| gender: non-binary \| nationality: Costa Rican \| birthdate: 19 May 1944 \| birthplace: Grecia, Puntarenas, Costa Rica \| deathdate: July 11, 1997 \| deathplace: Alajuela, Costa Rica \| deathcause: influenza \| restingplace: Grecia, Costa Rica \| almamater: UC San Diego, majored in Philosophy \| education: PhD in Philosophy \| occupation: essayist, playwright, director, and professor \| language: Spanish, also speaks English \| genre: drama, satire, realism \| awards: Premio Nacional Aquileo J. Echeverria, 1984; Costa Rica's Aquis Society Prize, 1986 \| partner: Margarita Brenes \| children: Aurelion, Raoul, Nairobi, and Maria Elena | Alvaro Ochoa (19 May 1944 - 11 July 1997) was a Costa Rican playwright, essayist, director, and professor. Ochoa was born in Grecia,Puntarenas, Costa Rica. He received a PhD in Philosophy from UC San Diego. They were awarded the Premio Nacional Aquileo J. Echeverria, in 1984, and Costa Rica's Aquis Society Prize, in 1986. They and their partner Margarita Brenes had four children. Their genres were drama, satire, realism. Ochoa died in Alajuela, Costa Rica, on July 11, 1997 due to influenza and was laid to rest in Grecia, Costa Rica. |
| name: Franz Beckmann \| gender: Male \| nationality: German \| birthdate: 12 April 1963 \| birthplace: Mannheim \| deathdate: 1 November 1991 \| deathplace: Nuremberg \| deathcause: heart attack \| restingplace: cemetery in Nuremberg \| almamater: University of Mannheim \| education: PhD in modern German literature with a special focus on Franz Kafka and his influence on modern German literature \| occupation: writer of academic textbooks on Kafka, literary essays and a semi-autobiographical novel inspired by Kafka and his novella The Metamorphosis \| notableworks: Kafka – Myth and Transformation, Kafka and his Country, A Novel on my Life. \| language: German \| genre: literary criticism, literary scholarship, fiction \| awards: Kafka Prize for his first novel, A Novel on My Life, and the Franz Kafka Prize for contribution to Kafka studies \| mother: Martha Beckmann \| father: Heinrich Beckmann \| partner: Hans von Holthusen \| children: none | Beckmann was a German writer. Beckmann was born on April 12 1963 in Mannheim to Martha and Heinrich Beckmann. He attended the University of Mannheim, where he earned a PhD in modern German literature with a special focus on Franz Kafka and his influence on modern German literature. Beckmann received Kafka Prize for his first novel, A Novel on My Life, and the Franz Kafka Prize for contribution to Kafka studies. Beckmann died in Nuremberg on November 1, 1991 of a heart attack. He was buried in Nuremberg. Beckmann and his partner Hans von Holthusen had no children. |

Table 9: The individual notability types included in SynthBio. The 'Rank' column indicates the rank of each type in terms of number of individuals in Wikipedia belonging to that type. Negative rank values indicate reverse rank, e.g., *mountaineer* is the third least common type.

| Type | Attributes | Rank |
|---|---|---|
| musical artist | instrument, genre, hometown, citizenship, education, years active, label, associated acts, awards | 2 |
| sportsperson | sport, country, hometown, citizenship, education, collegeteam, event, position, years active, retired, height, weight, coach, national team, worlds, olympics, paralympics | 3 |
| scientist | occupation, fields, known for, hometown, citizenship, alma mater, thesis title, thesis year, doctoral advisor, awards, institutions, notable students, influences, influenced | 4 |
| writer | alma mater, education, occupation, notable works, language, genre, awards | 6 |
| artist | known for, notable works, movement, alma mater, awards, elected | 10 |
| mountaineer | start age, notable ascents, final ascent, partnerships | -3 |
| spy | serviceyears, known for, criminal penalty, alma mater, occupation, codename, allegiance, agency, operation | -17 |
| theologian | alma mater, occupation, tradition movement, notable works, main interests | -15 |

## 8.2 Biography Generation Few-shot Examples

The follow are excerpts of the examples used in few-shot prompting. We show examples for two notability types: theologian and sportsperson.

```
theologian = {
  biography_examples: [{
    name: Ahmed Abdulhadi,
    attributes: gender: male | nationality: Jordanian | birth_date: 18 December 1936
        | birth_place: Madaba, Jordan | death_date: 20 August 1966 | death_place:
        Riyadh, Saudi Arabia | death_cause: car accident | resting_place: Amman,
        Jordan | alma_mater: American University of Beirut, Beirut, Lebanon |
        occupation: professor, writer, politician, Christian scholar |
        tradition_movement: Evangelical tradition within the Presbyterian Church in
        America | notable_works: "Disarmament as a Discipline for Peace," "The
        Peaceable Kingdom," and "Reason and Love of God" | main_interests:
        evangelism, pacifism, Christianity | mother: Nadia al-Hashimi | father:
        Husayn ibn Abdul Salam al-Hashimi (1916--1990) | partner: Zeina Khoury Al-
        Hashimi | children: Ali, Zeina, Rami, Laila,
    biography: Ahmed Abdulhadi (18 December 1936 - 20 August 1966) was a Jordanian
        professor, writer, politician, and Christian scholar. He was born in Mabada,
         Jordan to Ndia al-Hashimi and Husayn ibn Abdul Salam al-Hashimi. Abdulhabi
        attended the American University of Beirut. A member of of the Evangelical
        tradition, Abdulhabi\'s main interests were evangelism, pacifism, and
        Christianity. He is best remembered for his books "Disarmament as a
        Discipline for Peace," "The Peaceable Kingdom," and "Reason and Love of God".
         Abdulhabi died in a car accident in Riyadh, Saudi Arabia. He is buried in
        Amman, Jordan. Abdulhabi and his partner Zeina Khouri Al-Hashimi had four
        children: Ali, Zeina, Rami, and Laila.
  }, {
    name: Osborn Mobutu,
    attributes: gender: male | nationality: Congolese | birth_date: 26 December 1935
        | birth_place: Pointe Noire, Republic of the Congo | death_date: n/a |
        death_place: n/a | death_cause: n/a | resting_place: n/a | alma_mater: Yale
        University, Yale Divinity School, New Haven, Connecticut | occupation:
        theologian, clergyman, politician (past), political economist/philosopher (
        past) | tradition_movement: Judaism | main_interests: religion and non-
        Western countries, and how religion can adapt to other cultures | mother:
        Adela Deschamps | father: Charles Mobutu | partner: Yitzhak Goldstein |
        children: Jean-Luc Mobutu, Adele Mobutu, Charles Mobutu II,
    biography: Osborn Mobutu is a Jewish-Congolese theologian, clergyman, politician,
         politicial economist, and philosopher. Mobutu was born on December 26, 1935
         in Pointe Noire, Republic of the Congo to Adela Deschamps and Charles
        Mobutu. He attended Yale Divinity School in New Haven, Connecticut.
        Throughout his career Mobutu has been interested in religion and non-Western
         countries and how religion can adapt to other cultures. Mobutu and his
        partner Yitzhak Goldstein have three children: Jean-Luc Mobutu, Adele Mobutu
        , and Charles Mobutu II.
  }, {
    name: Wanisha Nurul Husna,
    attributes: gender: female | nationality: Indonesian | birth_date: 22 August
        1985 | birth_place: Bandung, West Java, Indonesia | death_date: 14 August
        2019 | death_cause: aneurysm | resting_place: Jakarta, Indonesia |
        alma_mater: Bandung Institute of Technology | occupation: theologian, writer
         | tradition_movement: Islamic feminism | notable_works: On the Nature of
        Women According to the Qur\'an and Islamic Tradition | main_interests:
        Islamic feminism, Islamic theology | mother: Rahmawati Nuur | father:
        Husnain Nurul Husna | partner: Nazaruddin Nadir | children: none,
    biography: Wanisha Nurul Husna (22 August 1985 - 14 August 2019) was an
        Indonesian theologian and writer, best remembered for her book "On the
        Nature of Women According to the Qur\'an and Islamic Tradition". She was
        born in Bandung, West Java, Indonesia and attended the Bandung Institute of
        Technology. Part of the Islamic feminism movement, Husna\'s main interests
        were Islamic feminism and Islamic theology. Husna died of an aneurysm in
        2019, and was laid to rest in Jakarta, Indonesia. Husna\'s parents were
```

```
                  Rahmawati Nuur and Husnain Nurul Husna. She and her partner Nazaruddin Nadir
                    had no children.
        }]
}

sportsperson = {
  biography_examples: [{
    name: Liu Wang,
    attributes: gender: female | nationality: Chinese | birth_date: 19 May 1931 |
        birth_place: Beijing, China | death_date: 25 January 2020 | death_place:
        Amsterdam, Netherlands | death_cause: breast cancer | resting_place: n/a |
        sport: speed skating | hometown: Beijing, China | citizenship: Chinese |
        collegeteam: n/a | event: short track speed skating | position: n/a |
        years_active: 1957-2003 | retired: 2003 | height: 5feet. 0inches | weight: n
        /a | coach: n/a | national_team: Chinese national speed skating team |
        worlds: 2 Silver, 3 Bronze | olympics: 1 Gold, 1 Silver, 1 Bronze |
        paralympics: n/a | mother: Liu Yun Qi | father: Liu Yan Zhen | partner: n/a
        | children: Liu Hong Feng,
    biography: Liu Wang (19 May 1931 to 25 January 2020) was a Chinese short track
        speed skater. Wang was born in Beijing, China on May 19, 1931 to Liu Yun Qi
        and Liu Yan Zhen. She began her speed skating career in 1957, and retired in
        2003. Wang computed with the Chinese national speed skating team and
        throughout her career has won 2 silver and 3 bronze medals at the world
        speed skating championship, as well as Olympic gold, silver, and bronze
        medals. Wang died of breast cancer while in Amsterdam, Netherlands on
        January 25, 2020. She is survived by her daughter Liu Hong Feng. She was 5
        feet tall.
  }, {
    name: Raaia Al Haji,
    attributes: gender: non-binary | nationality: Qatari | birth_date: 29 August
        1927 | birth_place: Doha, Qatar | death_date: 25 October 1999 | death_place:
        England | resting_place: East Grinstead | sport: cycling (mountain) |
        country: United Kingdom | hometown: Doha | citizenship: Qatar | education:
        Oxford University | collegeteam: Oxford University Cycling Club | event:
        cycling | position: n/a | years_active: 1947-53 | retired: 1953 | height: 5
        ft 8in | weight: 11st 0lb | coach: John Atkins | national_team: n/a | worlds
        : n/a | olympics: n/a | paralympics: n/a | mother: Janet Al Haji | father: n
        /a | partner: George Hickey (m. 1954) | children: Peter Hickey,
    biography: Raaia Al Haji (29 August 1927 - 25 October 1999) was a Qatari non-
        binary cyclist. Though Haji was born in Doha, Qatar and was a Qatari citizen
        , they competed for the United Kingdom. Their cycling career started in 1947
        - Haji competed with the Oxford University cycling club while a student at
        Oxford. Haji retired from the sport in 1953. Haji stood at 5ft, 8inches and
        weighed 11 stone. They were coached by John Atkins. Haji married their
        partner George Hickey in 1954 and they had one son: Peter. Haji died in
        England in 1999 and was laid to rest in East Grinstead.
  }, {
    name: Vinicio Silva,
    attributes: gender: non-binary | nationality: Brazilian | birth_date: 17 January
         1963 | birth_place: Sao Paulo, Brazil | death_date: n/a | death_place: n/a
        | death_cause: n/a | resting_place: n/a | sport: running | hometown: Sao
        Paulo, Brazil | citizenship: Brazil | education: University of Georgia
        (1972) | collegeteam: n/a | event: running | position: mid-long distance |
        years_active: 1976-1993, 1999-2001 | retired: n/a | height: 5ft 10in |
        weight: 145lb | coach: n/a | worlds: n/a | olympics: n/a | paralympics: n/a
        | mother: Maria dos Santos Souza | father: Manoel Silva,
    biography': Vinicio Silva is a Brazilian runner specializing in mid-long
        distances. They were born in Sao Paulo, Brazil to Maria dos Santos Souza and
         Manoel Silva on January 17, 1963. Silva attended the University of Georgia,
         graduating in 1972. Silva has been active competitively between 1976 and
        1993, as well as from 1999 to 2001. They are 5 feet, 10 inches tall and 145
        pounds.
  }]
}
```

### 8.3 Examples of Human Edits

For attribute lists - edits fell into three categories: corrections to (1) factual plausibility, (2) appropriateness, and (3) formatting. Though we did not collect statistics on the distribution of edits, we reviewed a sample and observed that most corrections were in categories (1) and (2). We include examples below.

Factual plausibility edits:

- **original:** death_date: 2054 **correction:** death_date: n/a

- **original:** mother: none **correction:** mother: Grace Madondo

- **original:** birth_date: 15 January 1913, death_date: 1 November, 1991, partner: Karl Bauer - 1839 - 1855 **correction:** partner: Karl Bauer - 1939 - 1955

Appropriateness edits:

- **original:** death_date: unknown - thought to have died in the 1970s **correction:** death_date: unknown

- **original:** death_cause: nothing is known, they seem to have simply disappeared **correction:** death_date: unknown

- **original:** children: none (though there is suspicion that Ayaksanis Zhaksylyk had a child before she met her husband) **correction:** children: none

Biography edits fell into three categories: corrections to (1) faithfulness, (2) fluency, and (3) formatting. We did not collect statistics on the distribution of edits. A manual review showed that most corrections were in category (1). We include examples below.

Faithfulness edits:

- **original:** ...Fritz von Lehmann is a non-binary author... **correction:** ...Fritz von Lehmann was a non-binary author...

- **original:** ... They have been diagnosed with multiple sclerosis... **correction:** [sentence deleted due to being unsupported by attribute list]

- **original:** Cruz died in 1935. **correction:** Cruz died on January 7, 1935 of throat cancer.

### 8.4 Inter-Annotator Agreement for Human Evaluation

Table 10 gives the inter-annotator agreement among three raters evaluating WikiBio and SynthBio for coverage and fluency.

## 9 Data Card

Our data card follows the template by McMillan-Major et al. [34].

**Dataset and Task Summary** The synthetic WikiBio benchmark is a multi-reference dataset for the evaluation of data-to-text generation systems trained on the WikiBio dataset [32]. It contains 2,249 list of attributes and 4,692 biographies. Unlike the original WikiBio, this dataset is fully synthetic and does not describe anyone who exists or existed. The dataset was created through a Human-AI

Table 10: Inter-annotator scores among three raters for human evaluation of WikiBio and SynthBio coverage and fluency properties.

|  | Coverage (Fleiss-Kappa) | Fluency (% of perfect agreement) |
|---|---|---|
| **WikiBio** | 0.4689 | 0.925 |
| **SynthBio** | 0.5214 | 0.95 |

collaborative approach where a large language model generates attributes and biographies, which are then refined by crowdworkers. Its intended use is as a clean corpus to evaluate these systems without memorization issues.

**Languages**    The dataset contains English text (BCP-47: `en`). Names and other attributes are often also transcribed in the respective local script.

## 9.1    Meta Information

**Dataset Curators**    The dataset was developed by researchers at Google Research with backgrounds in natural language processing and human-computer interaction.

**Licensing Information**    Apache 2.0

**Leaderboard**    There is no official leaderboard associated with this dataset and we discourage the use of the dataset as means to hill-climb automatic metrics.

## 9.2    Dataset Structure

**Data Instances**    We provide examples of the dataset in Table 8.

**Data Fields**    The dataset is distributed as a `json` file containing a list of instances with the following fields:

`notable_type`: the class of notable person this example describes following the categories in Table 9; `attrs`: A mapping from attribute to its value, e.g. "name: Hugo"; `biographies`: A list of strings; `ids`: A list of IDs that uniquely identify each biography; `serialized_attrs`: A linearized version of the attributes formatted in the same way as the original WikiBio dataset which can be used as input to a seq2seq model.

**Data Statistics**    SynthBio does not contain any splits – It can be used as a high-quality finetuning set [49] or, as explored in this paper, as a monolithic additional evaluation set.

The dataset has 2,249 infoboxes and between one and six references for each. On average, each infobox has 2.1 associated biographies (4,692 total). Table 1 shows additional data statistics.

## 9.3    Dataset Creation

**Curation Rationale**    SynthBio was created as a case study and feasibility test of human-AI collaborative corpus creation. We were interested in studying the limitations and opportunities of using a large language model to create data, how we can design the dataset to follow desired distributions (in our case gender, nationality, and notability type), and whether the associated challenges can be overcome through additional crowdsourcing steps.

The reason that WikiBio in particular was chosen as a target is that its limitations are well-studied in regard to non-attributable statements in references of generation datasets [e.g., 51, 19, 56]. It is further an interesting target since large models are typically trained on Wikipedia itself, which means that generations are not attributable to the model or the data, which is demonstrated by our experiments shown in Table 2. The new set can thus be a valuable resource to evaluate models on without suffering from hallucination or memorization problems.

**Communicative Goal**    The communicative goal that spans both the WikiBio and SynthBio task is to generate a multi-sentence biography of a person, grounded in a list of key-value attribute pairs given in the input. The biography should not contain any extraneous information, and faithfully cover all of the attributes in the input.

**Source Data: Initial Data Collection and Normalization**

**Source Data: Who are the source language producers?**  The initial few-shot examples were produced by the author of this work A.Y., while the additional attribute conditions were produced by all authors after consulting multiple experts in the research area of responsible AI. The actual initial dataset was then generated by a 137B parameter Transformer-based language model trained on public web documents.

**Annotations: Annotation process**  Annotations for SynthBio were carried out in a multi-stage process. First, a language model synthesized drafts of attribute lists describing fictional individuals. These were given to paid raters who edited the lists for factual plausibility, appropriateness, and formatting. Quality assurance was performed periodically on the ratings, with edge cases sent to the paper authors to adjudicate. The resulting attribute lists were used to synthesize drafts of natural language biographies. These biographies were then edited by raters for faithfulness (with respect to the attribute lists from which they were derived), appropriateness, and formatting. Raters had the option to discard biographies that could not be edited to meet these criteria within 2 minutes. Again, edge cases were adjudicated by the paper authors.

An in-depth discussion of the annotation process can be found in section 3 of the paper. The exact instructions given to the annotators as well as screenshots of the annotation UI are provided in the supplemental materials to this paper.

**Annotations: Who are the annotators?**  Annotators were paid full-time raters fluent in English from Hyderabad, India (thus their native dialect of English may differ from that of U.S.-based English speakers). A total of 13 annotators were involved in creating SynthBio.

**Personal and Sensitive Information**  We tried to ensure that none of the people described in the dataset actually exist or resemble people that exist. Since the design of the attributes follows those present in WikiData, only non-sensitive information is present in the attributes and associated biographies (Data that for a real person would be readily available). Any resemblance to actual persons, living or dead, is unintended and purely coincidental.

## 9.4   Considerations for Using the Data

Recapitulating the discussion in Section 6, we acknowledge that the data may contain spurious correlations that were not controlled and result from creating the seed data with a model. In addition, while the geographic diversity is much higher than in the original WikiBio dataset, the dataset does not cover all notability types or countries and does not describe real events that people from each country lived through. The data we create may thus not be reflective of the real tail of the test distribution. In addition, the crowdworkers themselves may introduce unconcious biases into the data that are not representative of the entire population covered in this dataset.

We conducted a fairness audit of the SynthBio dataset by testing a small sample of bios across a cross-section of social and cultural identities, and observed several potential fairness concerns, specifically, inconsistencies in gender identity and an overrepresentation of Western attributes. Bios often showed inconsistencies between a subject's gender identity and their pronouns, swapping between they/them pronouns and she/her or he/him pronouns. This was especially prevalent for non-binary subjects. For example, for the non-binary individual Susanne Strasser, the bio states: "Susanne studied at the Vienna Academy of Fine Arts and they were well-known for her feminist stance," switching pronoun usage within the same sentence. We note that these inconsistencies were observed in the final dataset, whose samples were audited by human raters specifically instructed to correct pronoun inconsistencies, suggesting stricter oversight may be required to address this issue in future data collection work. Additionally, there is an overrepresentation of Western institutions, awards, and attributes throughout the bios. For example, though the individual Seung Ki Ahn's nationality is Korean, he is described as serving in the US military. Additionally, for Olympian Beata Gogia, the infobox and bio dictate that her nationality, hometown, and citizenship are Georgian, but she represents the US in the Olympics. Though in isolation these issues would be innocuous, they occurred with enough frequency to constitute a potential fairness concern. These factors should be considered when using the dataset, especially in training or testing novel models.

Finally, if this dataset is used as an evaluation dataset, we also still encourage the use of the real dataset (possibly after cleaning and/or filtering it), to have numbers that are representative of both styles of dataset.

**Social Impact of the Dataset**    SynthBio and its creation technique helps further the research on the creation of evaluation suites, or challenge sets, that have highly controlled properties to test specific aspects of a model. In our case, SynthBio is designed to as uniformly as possible reflect people of different notability type, nationality, and gender such that each of these equally contributes to overall performance numbers. This design has the potential to improve the inclusivity of performance testing compared to random splits.

**Impact on Underserved Communities**    This dataset is only in English, which is the language with the most existing resources. It does, however, improve the representation of people from different countries and with different genders. It is one of the first, if not the first, datasets with a significant representation of gender-neutral references.

