# OpenReview forum: "SynthBio: A Case Study in Faster Curation of Text Datasets"
_NeurIPS.cc/2021/Track/Datasets_and_Benchmarks/Round2 — NeurIPS 2021 Datasets and Benchmarks Track (Round 2)_

### Official Review · Reviewer_ez3d · 2021-09-18
**The subject requires more coherent and in-depth research**

**Rating:** 7
**Confidence:** 3
**Clarity:** OK.

**Strengths:**

- The goal of reducing bias in training is of importance.
- The study has performed human annotations.
- The paper is well-written.

**Weaknesses:**

These points are not sorted in any way:

1. Let's say that the point of equalizing the numbers (in this paper) is only to help to create more accurate machine learning models via a better training dataset. In the paper, the numbers are considered equal to help reduce bias in training (such as #spies==#artists). This (and many more apparent equalities) does not reflect the world as is and can be argued that it would create worse models in training. For example, the number of males and females and non-binaries are all considered equal in the unedited version (and close after human editing) for all occupations. In an occupation, such as mining, where almost all of the subjects are male, using equal numbers would only mislead the model by removing an important piece of information.

2. The data should reflect reality and if there is a problem or bias in training, it should be handled through various modeling or pre-processing steps. I believe that the data should remain loyal without removing important information or statistics. For example, WikioBio contains 72,831 subjects in the train part, while the novel dataset only includes 2,793. It would be easy to subsample the former to create an unbiased dataset without the need for human annotations. Maybe another case study would have been a stronger choice and would better reflect the results of the proposed workflow.

3. In Figure 2, the distribution of people from the novel dataset is compared with WikioBio according to the world population. The English Wikipedia does not have to reflect the statistics of all populations. For example, there are fewer cabdrivers compared to politicians as famous people, while they are much more in real life. However, if the goal was only to create equal numbers, then there should be equal numbers from each country, which would raise other problems (e.g. #china==#guam is simply biased in sampling, and more problems as stated before).

4. The work argues that equal numbers would lead to better results, but it clearly will not. For example, the number of people from India has equal numbers in the dataset to a very small country. This if this country is split into two or more countries, the number of samples would be multiplied. This means that the same people are represented with more attention, even though nothing has changed. This is more evident in occupations where there are overlapping notability groups, such as "sportsperson", "swimmer", "rugby biography", ... . Perhaps it would be meaningful to find the right method for subsampling, as it would lead to better evaluation results.

5. One very important problem with the current paper is that it assumes that the Wikipedia infobox is complete, cohesive, and accurate, while the biographies are incorrect and should be based on the structured infobox. However, it can be argued (and easily showed) that many of the biographies are continuously edited from valuable sources, while the structured infoboxes in Wikipedia are usually handled by fewer users based on the updated biographies. Thus the base of the paper assumption for the case study is wrong (or unproven).

6. Continuing with notability groups of Appendix 1, there are several overlapping or duplicate notability groups, such as "sportsperson", "swimmer", "rugby biography", ... . In addition, they use different granularity levels (e.g. "college football player" vs "artist" or "officeholder"). Finally, many occupations are missing from the base Wikipedia pages (such as judges and soldiers). This also clearly shows that the infobox can be incorrect on many levels. Further study is required for other properties.

7. Another important point is the way the work and previous works have been evaluated. All works are evaluated based on the evaluation part of the novel dataset. As stated, the current dataset does not reflect statistics of the real world, famous people, or famous people submitted at Wikipedia, thus it seems incorrect.

8. Since the work uses manual editing by human annotators, the problems regarding bias in submitting could still persist. For example, if most annotators are male or with a computer science background or from the USA.

9. The change in model evaluation results could come from several steps: 1) synthetic generation, 2) equal numbers, 3) changing statistics, 4) human annotation, 5) dataset size. I would suggest changing the case study to a less complicated one where you can accurately analyze the impact of each part.


A few points regarding bias in Wikipedia, WikioBio, ... :
1. Wikipedia reflects famous people of the world, not a fair subsample of all populations. By design, it only includes people based on their popularity, not their impact or goodness. Thus the question is, does your work tries to create a fair method for the whole population or only for famous people? If the former, then approaches such as WikioBio cannot be compared in statistics as the goal is different. If the latter, then works such as WikioBio are much closer to the reality of famous people (as collected by the populations) than a synthetic dataset based on what the world statistics should be based on your view of life. Based on equal numbers taken for occupation (e.g. number of spies are equal to the number of researchers in the new dataset) and gender (male/female/non-binaries) in the new dataset, the novel dataset statistics don't reflect the current world or the world that should be. Thus I would not recommend comparing this work with works based on real data.

2. If the point is to show the existing inequalities or biases in Wikipedia subjects or the real world, there are too many existing open problems or specific questions about the current approach or the reality of the world that would be impractical to answer in such a short paper. The first question is, does inequal numbers mean bias.
3. It is true that Wikipedia may not reflect a 100% accurate view of all famous people. As it is mentioned in the paper, most subjects are from the EU and the USA. However, it is not clear how much of this apparent inequality comes from the real world view and how much is from the WIKI contributors. For example, there are many more male researchers and politicians in the world compared to women; thus reflecting this in Wikipedia or related datasets is not a weakness on their parts. As another example, even though there are fewer female subjects in soccer than males, while they are probably close in number in the real world, female sports make much less money than males in reality (through ads, tickets, ...), indicating less popularity of the subjects.
4. It can be agreed or argued that an actress or player from the USA or EU has more chance to be known in other countries (thus passes the notability test) than the opposite. The impact of this point (similar to others) requires separate studies and cannot be identified or resolved in a short unrelated paper.

5. It should be noted that the source of most studies is English Wikipedia. Thus, it would be more meaningful to include more subjects from English-speaking countries and close countries. If Wikipedia subjects from all languages are united, then the result could be fairer (needs investigation).

**Additional Feedback:**

The notes are given humbly to improve the work. With its primary goal, it can be improved into a strong and influential study. I suggest changing the case study to a less complicated one where you can accurately analyze the impact of your work.

**Correctness:**

Besides the previously mentioned points:

- This sentence from Introduction needs a source or argument: "Progress in machine learning depends on the availability of high-quality benchmark datasets, yet there is a shortage of such datasets for natural language generation"

**Documentation:**

It seems fine. However, it lacks information on the number or method of human annotators.

**Ethics:**

- The paper changes the statistics of famous people without proper consideration. For example, there is an equal number of people from different countries or occupations. This may lead to incorrect plots, information, etc.
- The generated names and occupations could coincide with real-world people and cause problems. For example, the fictional "Raymond Reddington" is a Russian spy.

**Relation To Prior Work:**

OK.

**Summary And Contributions:**

The paper mainly proposes a workflow in which human raters collaborate with a large language model to efficiently curate a high-quality labeled text dataset. The study applies the workflow to create a case study in curating a new structure-to-text dataset, SynthBio. The main technical point is the use of editing instead of creating new instances, which is faster and more accurate. And the authors argue that this method will reduce bias in training which is the overall goal of the workflow.

While the goal of reducing bias in training is of importance and the study has performed various human annotations, there are serious problems in its foundations, especially regarding bias studies, that make the result and the overall workflow unacceptable. As the main two results of the paper, it is stated in the abstract that "We show that our dataset of fictional biographies is less noisy than WikiBio, and also more balanced with respect to gender and nationality".

---

> ### Author Response · Authors · 2021-09-29
> **Thank you for your review.**
>
> Thank you for taking the time to write such a detailed review.
>
> One of the reviewer's primary concerns is that SynthBio modifies the distribution of various attributes so that they differ from "the real world." However, there has been quite a bit of research done showing that Wikipedia itself does not reflect the real world and contains ingrained biases. See ["It’s a Man’s Wikipedia? Assessing Gender Inequality in an Online Encyclopedia"](https://arxiv.org/abs/1501.06307), ["Controlled Analyses of Social Biases in Wikipedia Bios"](https://arxiv.org/abs/2101.00078), and [“First Women, Second Sex: Gender Bias in Wikipedia.”](https://dl.acm.org/doi/10.1145/2700171.2791036) While the reviewer is correct that taxi drivers are less likely to be considered notable enough for a Wikipedia page than politicians, it is not our intention to trick the model with completely improbable synthetic people. Instead, we argue that it is important for a structured text -> natural language system to be equally effective at generating a faithful biography for a black British female notable person as a white American male notable person, even if there are more of the latter present in the training set. We make the case that SynthBio evaluates these less likely but important-to-get-right cases in a way that real Wikipedia data doesn't. We will add a version of this justification to the paper text.

---

> ### Author Response · Authors · 2021-09-29
> **Response to list of weaknesses.**
>
> Responding point-by-point to the reviewer's concerns:
>
> 1. Just because there are more male miners than female miners doesn't mean a bio-generation system should be incapable of correctly generating a bio for a female miner. That being said, we agree that careful deliberation on desired biases would be needed before applying our method to generating training data. This is one reason why we focus solely on evaluation data in our paper.
> 2. We agree that subsampling would be another good method to create less biased evaluation data. However, there are a few limitations with this. First, you can only sample from what is already there. For some combinations of attributes, there is little to no data. Second, subsampled evaluation datasets still have the problem that the eval set contains only real people. A pre-trained model might have seen content about this person and memorized it. Third, real data is noisy. A biography might contain more or fewer details than the ones in its attribute box.
> 3. The point of Figure 2 is to show that we can control the distribution of places of origin in the synthesized data and make it somewhat more reflective of the real distribution of population in the United States. Our success suggests that it would also be possible to use our method to create data with some other desired distribution of places of origin.
> 4. Our work does not argue that equal numbers will lead to better results. Indeed, we don't propose any changes to the WikiBio training set, and we didn't expect models to perform better on SynthBio than on the original WikiBio validation set. We agree with you that we could have improved our method for sampling country of origin attributes to make it more representative of the real-world distribution of people.
> 5. Our work does not assume that the Wikipedia infobox is complete, cohesive, and accurate. We do make the assumption that in a structured data to natural text generation task, the natural text should contain the contents of the structured data and no extra information. Real-world Wikipedia very often does not meet that requirement, and we design SynthBio to try and meet it. Because real Wikipedia is noisy in the ways you describe, SynthBio is a better dataset for evaluating specifically whether natural language can be produced that is faithful to structured data without any hallucination. One additional experiment we could have run would have been to take SynthBio and randomly drop out attributes so as to mimic the noisy data seen in the real world.
> 6. The problem of overlapping or duplicate notability groups in real Wikipedia stems from the fact that this data was produced ad-hoc by thousands of Wikipedia contributors. For SynthBio, we limited the dataset to eight notability types which we judged to be at a similar granularity and roughly disjoint. These were: musical artist, sportsperson, scientist, writer, artist, mountaineer, spy, and theologian. One of our goals was to create an evaluation dataset that was less likely to be noisy and incorrect than real WikiBio data.
> 7. In Table 7, we evaluate on both the original WikiBio eval set and SynthBio. We don't argue that SynthBio should replace the WikiBio validation set for evaluation. Rather, we argue that SynthBio is useful for augmenting the real-world validation data as it evaluates on cases which are important to get right but may not be common in real Wikipedia-derived data.
> 8.You are absolutely correct with this. Specifically, our annotators were all based in India, which certainly influenced the cultural references and historical events they were familiar with. We will expand the last paragraph in Section 6 to discuss the potential biases brought on by choice of annotators.
> 9. This is a good suggestion, and future work on disentangling these effects will be important.

---

> ### Author Response · Authors · 2021-09-29
> **Responding to reviewer's points on bias in Wikipedia.**
>
> 1. To be clear, our method is a way to create synthetic data which differs from real data in desirable ways. It is up to the implementer to decide how to define "desirable." In our case, we use our proposed method to create SynthBio, which aims to address the problems of train/test leakage, gender bias, and location bias. We do not attempt to address Wikipedia's bias toward famous people.
> 2. Unequal numbers does not necessarily imply bias, although sometimes it can (see papers on Wikipedia linked below). The goal of our paper was not to study the existing inequalities and biases present in Wikipedia, as this is a very well-studied area that indeed goes well beyond the purview of this paper. Rather, our paper makes the claim that there exist biases that should not influence the performance of a structured text to natural language generation system, and that our proposed dataset addresses some of these biases.
> 3. We refer the reviewer to the papers analyzing Wikipedia listed below.
> 4. The point of our paper was not to analyze the properties of Wikipedia. There has been a ton of great research done in this area, and we will add more citations to it into our paper.
> 5. The question is: just because English Wikipedia contains an overrepresentation of people from English-speaking countries, should a method trained to output English biographies given English infoboxes be better at this task for personages from English-speaking countries than for personages from other countries? We argue that the answer to this should be no.

---

### Official Review · Reviewer_YubP · 2021-09-18
**Interesting dataset generation method with a few limitations**

**Rating:** 7
**Confidence:** 4

**Strengths:**

The paper presents an interesting methodology for human-AI collaborative dataset generation which should be appealing to many researchers in the field trying to address the problems of memorisation and bias present in modern LMs. The authors take great care to address the ethical concerns arising both from the data and the dataset generation process.

**Weaknesses:**

The SynthBio dataset is too small to be used for training (so that its attractive properties can be used directly by the models). In addition, the process described in this work only applies to synthetic data generation which might be a limitation for some researchers. The biggest limitation however is that the main motivation factors for this work haven't been fully explored or aren't fully supported by the experiments. The details are in the correctness and additional feedback sections.

**Additional Feedback:**

The main stated reasons for creating SynthBio (in comparison to WikiBio) is a) to sidestep the problem of memorisation of large LMs and b) to control for noise and biases of naturalistic data. These should be weighed against the cost of creating the dataset (and tradeoffs between size and quality) as opposed to e.g. editing/augmenting the original WikiBio. This is the first missing comparison: what if instead of LM-generated attributes/bios, the annotators corrected the original Wikipedia bios? This of course would leave the problems of skewed attribute distributions, but that would be possible to account for with different test splits each with different criteria (geography, gender, occupation). Beyond the comparison to editing WikiBio, it is interesting that the results in section 4 (lines 205-210) suggest that these biases of WikiBio don't seem to affect the performance of the trained models w.r.t these attributes (of course it's still a useful diagnostic). Finally, regarding the element of memorisation, there is a missing comparison with a non-fine-tuned version of T5. This would demonstrate how much of the performance on WikiBio is due to pure memorisation (the kind shown in Table 2) compared to memorisation due to overfitting.

Regarding the edits done by the annotators, it would be good to get an idea of the distribution of edit types (deleting hallucinated information, adding missing information, fixing grammar or typos, etc).

**Clarity:**

The paper is generally well written but there are a number of minor issues/typos:

- The tables and figures should be closer to the text that references them.
- Line 9: "comprised of" -> "composed of" or "comprising" (without of)
- Add a reference to Tables 3 and Figure 2 in lines 34-5 to match the reference to the statistics of Table 1.
- Figure 1 has multiple references to terms and methods that are not defined until later in the paper (few-shot examples, LLM, LLM-Dialog) - I really like the shape/colour coding of the stages though.
- Line 45: "is scarce" -> "are scarce"
- Table 2: it's not clear what the non-highlighted bits of text are? Are they all inaccurate (hallucinations)? If so, I would rephrase the "somewhat accurate" comment in the caption.
- I don't think there "infinite ways" to express a list of attributes in natural language (line 72).
- Line 122: According Figure 1 and previous description in the text, the model is not queried for an individual's nationality, gender and birthdate; they are sampled programmatically (step 1b)
- Appendices:
 - Circular reference of Appendix A.1 in Appendix A.1 (line 592)
 - Licensing information marked as TODO (line 741)
 - Missing Table reference on line 745

**Correctness:**

The dataset generation process is for the most part sound and well designed. The details presented in the paper and the appendices are very thorough. There are some areas that further clarification is needed though. First, the reported perplexity of the bios in Table 1 seems to indicate that the unedited biographies are by far the most fluent whereas the real Wikipedia (human-authored) ones are the least fluent (and the edits of the synthetic bios make them less fluent). These counterintuitive results seem to point to the fact that PPL as measured by GPT-2 is very biased towards LM-generated text. The other, and more important point concerns the attribute distribution. While the construction method allowed for a balanced distribution of all attributes (types, gender, ethnicity etc. could have been controlled for at the prompt stage) and the authors explicitly call out that they ensured an equal gender representation (line 68), it's not clear why there are still imbalances in the distributions as it is evident from Table 1 and Figure 2. There is also lack of reporting around the quality control during the dataset annotation process. The authors note in Appendix B.3 that quality assurance was "performed periodically on the ratings" and that there was a process of adjudication. However, there is no mention of inter-annotator agreement (or adjudication overturn rate if there a single annotator per example), nor of QA during the editing stage.

There are also a few issues/questions with the examples in Table 7 (Appendix A).
- Are these examples from the edited or unedited version of the dataset?
- In all of the examples, the education is marked as PhD - while I appreciate the idea that people from all genders, countries and times can get a PhD, I am hoping that the full dataset contains a more realistic distribution of educational levels.
- The gender attribute for the last example is capitalised ("Male"); this might be a typo (by the LM?) but it made me wonder whether spelling differences or typos can affect the quality of the produced bios.
- Also on the same example, the generated bio first sentence refers to the entity by their last name only. Is this a typical variant in the generated text? I can see how it might lead to reference ambiguities downstream.
- In the first example, the fact that Tacettin Guntekin's notable works are books isn't mentioned in the attribute list. Is this a type of an allowed inference on the part of the model (if this is an edited example)?
- In the third example, there is a missing space after Grecia, but more importantly, the first preposition is "He" rather than "They". Is this an uncaught error by the model (if this is an edited example)?

**Documentation:**

Yes.

**Ethics:**

No additional concerns.

**Relation To Prior Work:**

Yes, there is a discussion on how the proposed approach and dataset differ from similar works in curated data-to-text dataset, synthetic dataset generation and human-AI collaborative dataset generation.

**Summary And Contributions:**

The paper describes the creation of SynthBio, a synthetic dataset of biographies, serving as a companion to the WikiBio dataset. The authors use an AI-human collaborative approach to the creation where a large-scale LM is asked to generate attributes for fictional humans (similar to Wikipedia infoboxes) and biographies based on those attributes and a human annotator is asked to rate and edit both. The resulting dataset contains a more balanced set of attributes compared to WikiBio and in addition it avoids the problem of using real human biographies which can be memorised by large LMs. Both properties make SynthBio an attractive evaluation dataset for data-to-text generation systems.

---

> ### Author Response · Authors · 2021-09-29
> **Response to Reviewer YubP**
>
> > Paper lacks discussion of quality control methodology / inter-annotator agreement.
>
> During our periodic quality reviews, we reviewed samples of edits and indicated whether any should be overturned or adjusted. We have updated the supplementary materials to include documentation of these reviews for both the attribute and biography editing phases.
>
> For the attribute and biography editing phases, a single annotator was assigned to each item, thus we did not consider inter-annotator agreement. However, to ensure adherence to our provided guidelines, each annotator went through a training phase during which we provided feedback on initial annotated data and answered questions. We will elaborate this point in the paper.
>
> For human evaluation of the final dataset, two annotators were assigned to each item, but we only reported the average scores across all annotations in Table 4. We will add agreement numbers here.
>
> > The SynthBio dataset is too small to be used for training (so that its attractive properties can be used directly by the models).
>
> We cited the related paper “[Process for Adapting Language Models to Society (PALMS) with Values-Targeted Datasets](https://arxiv.org/abs/2106.10328)” in our data card which shows that a very small high-quality dataset can actually be during a second finetuning stage. However, we did not experiment with this approach and instead focused on the evaluation qualities of SynthBio. Since our construction methodology requires human oversight, its application to the construction of large-scale datasets is limited and we intend to explore in future work how to utilize it to build larger corpora.
>
> > These counterintuitive results seem to point to the fact that PPL as measured by GPT-2 is very biased towards LM-generated text.
>
> This is consistent with findings from prior work (e.g., [this paper]((https://aclanthology.org/P19-3019/)) and is another reason why we chose to add the post-editing stage which leads to a PPL much more consistent with the original data. Of course, the distribution will not be the same – however, when evaluating with PARENT, the focus is much more on what is being said rather than how it is being said which alleviates the issue.
>
> > It's not clear why there are still imbalances in the distributions as it is evident from Table 1 and Figure 2.
>
> Table 1: As shown in the table, the language model struggles to produce “they” references which all have to be added by the annotators. As pointed out in the comments about Table 7, this is a noisy and imperfect process. In future work, we will add additional quality filters to catch uncorrected pronouns.
> Figure 2: There are two potential sources of this non-uniformity: (1) We sampled countries and not continents which means that there is a natural bias toward continents with more countries. (2) Our filtering process may have led to a higher rejection rate for certain countries. We will further investigate reason (2) and add a statement in the discussion that sampling by country leads to unequal distribution over continents which may or may not be desired.

---

> > ### Author Response · Authors · 2021-09-29
> > **Response to Reviewer YubP continued**
> >
> >
> > Thank you for your careful readthrough of the non-cherry-picked qualitative examples in Table 7.
> >
> > > Are these examples from the edited or unedited version of the dataset?
> >
> > All examples are from the final version of the dataset, after all edits. We will clarify this in the caption.
> >
> > > In all of the examples, the education is marked as PhD - while I appreciate the idea that people from all genders, countries and times can get a PhD, I am hoping that the full dataset contains a more realistic distribution of educational levels.
> >
> > This is a good point. Education level is not one of the attributes we controlled for. We just checked and found that 5.1% of SynthBio examples have "PhD" listed in their education attribute, which is indeed larger than the US population's level (2%). We will incorporate this finding in our discussion of biases in Section 6.
> >
> > > The gender attribute for the last example is capitalised ("Male"); this might be a typo (by the LM?) but it made me wonder whether spelling differences or typos can affect the quality of the produced bios.
> >
> > This comment is very helpful and we will bring it up in our discussion section. We currently only state “we note that for any dataset created fully or in part by human workers, the final dataset is only as good as the skills of the humans involved.” We will expand this argument to also incorporate a discussion of noise in the data.
> >
> > > Also on the same example, the generated bio first sentence refers to the entity by their last name only. Is this a typical variant in the generated text? I can see how it might lead to reference ambiguities downstream.
> >
> > Another good point. This is likely a product of the fact that we aren't using an LM that was finetuned on Wikipedia, and while all Wikipedia bios ought to start with the full name of the individual, this may not be true for biographies in the LM's training set at large.
> >
> > > In the first example, the fact that Tacettin Guntekin's notable works are books isn't mentioned in the attribute list. Is this a type of an allowed inference on the part of the model (if this is an edited example)?
> >
> > We did not ask our annotators to go to this level of subtlety. Since Tacettin's occupation is listed as novelist, it is reasonable to assume that his notable works are books.
> >
> > > In the third example, there is a missing space after Grecia, but more importantly, the first preposition is "He" rather than "They". Is this an uncaught error by the model (if this is an edited example)?
> >
> > The missing space after Grecia was most likely introduced by the annotators. You are also correct that the annotators should have replaced "he" with "they," but they missed this. These show that our method is not perfect, and final dataset quality is still strongly influenced by the abilities of the annotators. We will extend the discussion of these examples in the paper.

---

### Official Review · Reviewer_T9FG · 2021-09-20
**Paper presenting desirable traits for NLG evaluation alongside concerns in dataset construction.**

**Rating:** 7
**Confidence:** 4
**Clarity:** The paper is largely well written.

**Strengths:**

- The dataset presents improved reference texts in comparison to WikiBio which will be of value to language generation evaluation.

- The synthetic nature of the dataset allows for control over distribution of individuals over attributes in the dataset which helps control for the biases present in datasets such as WikiBio. The synthetic nature also allows evaluation under a setup where models havent memorized biographies of individuals.

- The paper also provides some novelty in the method of dataset construction for text generation in being generated by human-edits atop generations of an LLM.

**Weaknesses:**

1. While the paper indicates that SynthBio is of high quality in the eyes of human raters, the paper does not present an analysis of two important aspects of the dataset: the effects of edits made by humans to the dataset and any possible biases in the dataset because of the way it was generated using large language models.

2. While the paper presents an approach to build a dataset for a specific structure-to-text application the claims made in the generality of the approach are somewhat unvalidated. For example, lines 25-28 present the method as applicable to a much broader set of datasets ("efﬁcient labeled text dataset curation") than what the paper validates, claiming generality to other language generation tasks would be the extent of what is reasonable. This is exacerbated by the reliance of the approach on large language models, infrastructure that only select research groups have access to. I would ask that the authors tone down claims of generality or provide arguments of reasonable ways the proposed method may be applied in diverse other contexts.

Elaborations regarding the first weakness:

The paper does not address an important dichotomy about the value of human edits vs the value of model pre-generating text. If we expect to see large edits to a models output one would need to question how much the model helps (in reducing human effort of authoring texts) and hurts (by introducing a bias where text is drawn from the models distribution alone) and if a cheaper model than an LLM may have been used. On the other hand, if the model text is largely unedited or presents trivial edits one must question this datasets utility (and of human evaluation as others have done given that current text generation models fool them easily: [Clark et al. ACL 2021](https://aclanthology.org/2021.acl-long.565/)) for measuring meaningful progress. A sweet spot between these two extremes is likely to exist which the paper does not address. The statistics of edit distances (Table 5) seem to be the one presentation of the effect of edits but these are somewhat hard to place in context for their meaning.

Q1: The supplementary materials detail three kinds of edits that annotators made to the attribute lists and generated bios. Were any statistics on the type of edits collected? Sharing a range of examples of edits made by humans would also be helpful.

Q2: While the annotator instructions asked annotators to not spend beyond 2 minutes on an edit the average time spent for the biography is about twice that. Does this imply that the text from the LLM required heavier edits? How would the number of edits required compare to text generated from a more broadly accessible model (eg. GPT-2)?

Some experiments which might shed more light on the model biases and effect of edits:

Q3: Did you attempt to perform evaluations (human and automatic of Section 4) on the unedited bios generated from an LLM? If one were to see similar performance as the human edited dataset the edits might seem somewhat unimportant?

Q4: Given that others ([Chintagunta et al NAACL-ws 2021](https://aclanthology.org/2021.nlpmc-1.9/), cited in the Related works) have shown it possible to use GPT3 text as a way to build an augmented summarization dataset, is it reasonable to say that one way to test if the text generated from the LLM is biased in some way would be to fine-tune a T5 (or other model) on unedited attribute lists and bios from the LLMs as augmented data followed by evaluation on SynthBio?

Q5: Were any attempts made to train classifiers of the kind presented in [Ippolito et al. ACL 2020](https://aclanthology.org/2020.acl-main.164.pdf), which shares authors with this submission, to check if trained classifiers were able to tag text in the edited vs unedited text of the bios as human vs machine written? If the edits made detection of machine text harder it would indicate the value of the human edits.

Of course, other experiments of the authors design could also answer these concerns.

**Additional Feedback:**

Other questions:
Lines 148-151: Questions about few-shot examples:

Q6: How were the few shot examples authored? Did humans generate attributes and bios from scratch, or was a different procedure followed? Please elaborate.

Q7: Were the same few shot examples used for all the biographies in the dataset for a given notability type?

Q8: Were any experiments conducted on the number of few shot examples needed for reasonable performance or on the effect of using different sets of few-shot examples to generate the bios?

Q9: Will all of the human authored few shot examples be included in the dataset release?

**Correctness:**

The dataset seems to be constructed in a sound manner, aside from the critiques raised above.

**Documentation:**

- The dataset was shared with reviewers but does not seem accessible to the public yet.

- Data Card says the license is TODO (L741) but other parts of the paper suggest that it is CC-BY (L516). This requires clarification.

- The paper does not seem to present a maintenance plan.

- Given that the dataset was built using a large language model, descriptions and details for this model would be important to include in this paper or in accompanying documentation. Lines 85-90 seem to be the only lines documenting these models, this is insufficient.


**Ethics:**

None.

**Relation To Prior Work:**

Related work is discussed with sufficient detail. But, the authors should make note to cite: [Fanton et al, ACL 2021](https://aclanthology.org/2021.acl-long.250.pdf) who present a dataset generated using GPT-2 in a editing loop with human annotators in the context of hate speech counter narrative generation.


**Summary And Contributions:**

The authors propose a structure-to-text dataset, SynthBio, built using structure and text generated from a large language model followed by edits made by human annotators. SynthBio presents a dataset of Wikipedia-infobox like attributes paired with multiple biographies tied to the set of attributes. The individuals in SynthBio are hypothetical individuals allowing the dataset to control for coverage over attributes (along dimensions such as gender, nationality, notability type etc.) and allowing model evaluation where memorizing training data wouldnt help performance on this dataset. The authors also human evaluate the quality of the SynthBio and show it to be of high quality in the eyes of human evaluators. Further they demonstrate that SynthBio also presents higher quality reference text for structure-to-text models compared to prior work in WikiBio.

---

> ### Author Response · Authors · 2021-09-29
> **Thank you reviewer T9FG.**
>
> ### Re: Paper lacks discussion of biases in the dataset due to its being generated by an LLM.
> We will include a more thorough discussion of LLM-introduced biases in the camera-ready. In section 6 (lines 279-297) we start to discuss the importance of human annotation in synthesis pipelines, exactly for the purpose of countering biases introduced by models. We noted that models have been observed to exhibit potentially harmful biases and pointed out a few examples of such bias, such as the fact that SynthBio contains more individuals whose birth place does not correspond to their nationality than WikiBio. We also emphasize the importance of research into controllable generation for improving the usefulness of model-assisted dataset curation (line 294).
>
> ### Re: Paper lacks discussion of what types of edits were made
> For attribute lists - edits fell into three categories: corrections to (1) factual plausibility, (2) appropriateness, and (3) formatting. Though we did not collect statistics on the distribution of edits, we reviewed a sample and observed that most corrections were in categories (1) and (2). We include examples below. We have also updated the supplementary materials to include a sample of edits.
>
> Factual plausibility edits:
> original: death_date: 2054 → correction: death_date: n/a
> original: mother: none → correction: mother: Grace Madondo
> original: birth_date: 15 January 1913, death_date: 1 November, 1991, partner: Karl Bauer - 1839 - 1855 → correction: partner: Karl Bauer - 1939 - 1955
>
> Appropriateness edits:
> original: death_date: unknown - thought to have died in the 1970s → correction: death_date: unknown
> original: death_cause: nothing is known, they seem to have simply disappeared → correction: death_date: unknown
> original: children: none (though there is suspicion that Ayaksanis Zhaksylyk had a child before she met her husband) → correction: children: none
>
> Biography edits fell into three categories: corrections to (1) faithfulness, (2) fluency, and (3) formatting. We did not collect statistics on the distribution of edits. A manual review showed that most corrections were in category (1). We include examples below. We have also updated the supplementary materials to include a sample of edits.
>
> Faithfulness edits:
> original: …Fritz von Lehmann is a non-binary author… → correction: …Fritz von Lehmann was a non-binary author…
> original: … They have been diagnosed with multiple sclerosis… → correction: [sentence deleted due to being unsupported by attribute list]
> original: Cruz died in 1935. → correction: Cruz died on January 7, 1935 of throat cancer.
>
> ### Re: Paper overstates generality of the method - it is only applicable to language generation tasks.
> We will emphasize that this is an area for future research, and that we hope to validate the human-AI collaborative approach to synthesize other types of text datasets. Reviewer t9fg also asks whether a cheaper model could have been used. We acknowledge that certain steps of the synthesis pipeline, such as the generation of particular attributes, could potentially have been achieved by a smaller model, although such models typically lack the ability to generate fluent text which was required for our task. An interesting line of future research would be developing model selection strategies for scaled-up synthesis efforts.
>
> ### Re: If edits were large, then does the pipeline reduce human effort? If small, then is the dataset useful?
> This is a great point and we will expand this discussion in the paper. In future experiments we’d like to compare the time humans spend editing synthesized examples versus writing them from scratch. This metric will enable us to characterize the quality vs quantity trade-offs in our pipeline. There is some evidence (Wu, 2021) suggesting that editing should be faster, but how much faster will be both task and model dependent.
>
> We note anecdotally in the paper that humans performed many small (in terms of edit distance) but high-impact edits on the generated dataset, e.g., many of the edits to biographies were changing pronouns to match the gender indicated by the infobox. This edit can be made quickly (the error is easily identified, only a few characters need to be changed), yet it significantly improves the faithfulness of the biography.
>
> In future data synthesis projects we will more rigorously track edit types by asking editors to label / categorize errors as they work.

---

> > ### Author Response · Authors · 2021-09-29
> > **Thank you reviewer T9FG - continued.**
> >
> > ### Re: Responses to individual questions under “Weaknesses”:
> >
> > Q1: Please refer to above response under “Paper lacks discussion of what types of edits were made”.
> >
> > Q2: A cost benefit analysis of model expense versus editing time would be a worthwhile study for future work. We will also investigate methods for worker selection and training to further reduce annotation cost. The costs observed for SynthBio are a baseline for future data synthesis efforts.
> >
> > Q3: We will add this comparison to the camera-ready. We anticipate the unedited generations will suffer from similar issues as the original set.
> >
> > Q4: This is a great direction for future research, and would provide an interesting additional measure of the impact of edits on the final dataset.
> >
> > Q5: We did not attempt to train classifiers to distinguish between edited and unedited text. This would also be an interesting experiment, although we anticipate there are many shortcuts a model could take for this task, e.g. learning that prevalence of “they” pronouns corresponds to edited text.
> >
> > Q6: Humans generated attributes and bios from scratch, based on exemplar pages provided in the official references for each schema (e.g. for sportsperson, the official reference https://en.wikipedia.org/wiki/Template:Infobox_sportsperson lists Kaori Icho’s page as an exemplar).
> >
> > Q7: Yes, three different examples were created for each notability type.
> >
> > Q8: We did not conduct this experiment, but it would be great for future work. We observed anecdotally that three examples were the minimum needed to generate diverse examples. We were also limited by the model’s context window of 900 tokens. We attempted to select few shot examples that were diverse in terms of attribute values represented.
> >
> > Q9: Yes, the human authored few shot examples will be included in the dataset release.
> >
> > We will also make sure to cite Fanton et al, ACL 2021 in the camera-ready.

---

> ### Comment · Reviewer_T9FG · 2021-09-29
> **Thank you for the responses - updated score.**
>
> Thank you for your responses, conditional on the changes and a few additional ones noted below I think this makes a good paper. My score has been updated appropriately.
> 1. Types of edits and examples - Thank you for adding some examples. I would recommend the examples (perhaps more than just the handful shared here) be added to the appendix with additional context. It is compelling to see the edits.
> 2. Over statement of some claims - I believe more general claims would be apt when the claims are validated in intended future work. I would ask that the claims be toned down, specially in the light of authors coming from large tech firms more likely to see broader discussion in the media than others. A change to the paper to prevent hype would be responsible, but I leave it up to the authors to choose as they see fit.
> 3. Documentation for the model used to synthesize the dataset will be nice to have given its importance to this work, the current documentation is insufficient. At the least: A more detailed description of training data (for LLM and importantly for LLM-DIALOG), expense of training, details of evaluation performed on any other datasets prior to use here, and any details on tuning the models for use in dataset synthesis.
> 4. The following promised additions will be nice to see in the camera-ready: 1) Large edits vs small edits and human effort. 2) Results of evaluation performed with unedited text from models (Q3).  3) Human authored few-shot examples to facilitate examination of their effects in future work (Q9). 4) The response to Q6 regarding authoring of few-shot examples.

---

### Official Review · Reviewer_gR2s · 2021-09-20
**A good contribution**

**Rating:** 7
**Confidence:** 4
**Correctness:** Yes, the paper is technically sound.
**Clarity:** Yes, the paper is very well-written.

**Strengths:**

(1) Overall, I think this is a good contribution with clear motivation, well-designed experiments, and in-depth analysis. The proposed dataset will undoubtedly benefit the research community.

(2) The paper is very well-written and easy to follow. The dataset construction process is described clearly, and I agree that human-AI collaboration is a promising direction for dataset construction.

(3) The newly proposed dataset addresses some of the issues in WikiBio which has been discussed in the paper.

(4)  I like the analysis part that explores the performance differences for the same models on the WikiBio dataset and SynthBio dataset using automatic metrics.

(5) I think the “Discussion and broader impact” part is very well-written and makes some points for the value of curating datasets and synthetic data.

**Weaknesses:**

The explorations on three potential reasons causing performance differences in Table 6 should take more pages. I think this is a very interesting part and should not be taken lightly.

**Additional Feedback:**

See the weaknesses.

**Documentation:**

Yes.

**Ethics:**

There are some ethical issues on whether or not datasets should be curated. And the paper makes in-depth discussions on this in its "Discussion and broader impact" part.

**Relation To Prior Work:**

Yes, prior works are discussed properly.

**Summary And Contributions:**

The paper proposes a curated dataset for structure-to-text tasks, consisting of infoboxes for fictional people paired with their biographies. The construction of this dataset includes both generative language models as well as human annotators.

---

> ### Author Response · Authors · 2021-09-29
> **Thank you reviewer gR2s.**
>
> Thank you for your thoughtful review. We will expand the discussion of Table 6 in the camera-ready version of our paper. For future work we will explore conducting more targeted experiments that would enable us to disentangle the different factors influencing the difference in model evaluation we observed between WikiBio and SynthBio.

---

### Author Response · Authors · 2021-09-29
**Thank you to the reviewers.**

We would like to thank all the reviewers for their thoughtful comments and look forward to incorporating reviewers’ suggestions for improvement. We’d like to use the additional page given for the camera-ready to expand discussion of important points identified by reviewers as requiring more attention, such as potential model and annotator-introduced bias in the final dataset. We also want to emphasize that our overall goals in creating SynthBio were (1) to demonstrate a novel methodology for dataset curation, and (2) to create an additional tool for model researchers in evaluation of structure-to-text models (rather than to supplant WikiBio). Our paper takes the position that researchers benefit from having the option to study the world as we want it to be, in addition to the world as it is (where “world” refers to extant sources of data which may contain biases).

---

### Decision · Program_Chairs · 2021-10-11

**Decision:**

Accept

**Comment:**

The reviewers agreed that this is an interesting idea that was clearly presented. There were a number of minor concerns from the reviewers regarding how AI-human collaboration affects dataset bias, but these were clarified in the discussion. Exciting and promising direction. I recommend it is accepted.